# Leveraging data from the Genomes-to-Fields Initiative to investigate genotype-by-environment interactions in maize in North America

Marco Lopez-Cruz [1,2] ✉, Fernando M. Aguate[1,2], Jacob D. Washburn[3], Natalia de Leon[4], Shawn M. Kaeppler[4,5], Dayane Cristina Lima [4], Ruijuan Tan[6], Addie Thompson [6,7], Laurence Willard De La Bretonne[4] & Gustavo de los Campos [1,2,8] ✉

Genotype-by-environment (G×E) interactions can significantly affect crop performance and stability. Investigating G×E requires extensive data sets with diverse cultivars tested over multiple locations and years. The Genomes-to-Fields (G2F) Initiative has tested maize hybrids in more than 130 year-locations in North America since 2014. Here, we curate and expand this data set by generating environmental covariates (using a crop model) for each of the trials. The resulting data set includes DNA genotypes and environmental data linked to more than 70,000 phenotypic records of grain yield and flowering traits for more than 4000 hybrids. We show how this valuable data set can serve as a benchmark in agricultural modeling and prediction, paving the way for countless G×E investigations in maize. We use multivariate analyses to characterize the data set's genetic and environmental structure, study the association of key environmental factors with traits, and provide benchmarks using genomic prediction models.

Genotype-by-environment (G×E) interactions play a major role in crop performance and stability[1,2]. Modeling and studying G×E has enjoyed a resurgence in recent years with both the development of new approaches, the increased availability of high throughput genotype and environmental data, and the increased interest in applying previously developed approaches on larger datasets and with more high-powered modern computers. These advances have been reviewed extensively elsewhere[3–5].

Numerous attempts have been made to use machine learning and deep learning approaches on this problem with some success, but also the continued realization that larger and more comprehensive data sets are likely needed to truly take advantage of deep learning approaches[6–9]. Another family of approaches that has shown great promise in recent years is the use of a traditional genomic best linear unbiased prediction (GBLUP) models but with the incorporation of environmental effects via environmental indices and other reduced

[1]Department of Epidemiology and Biostatistics, Michigan State University, East Lansing, MI 48824, USA. [2]Institute for Quantitative Health Science and Engineering, Michigan State University, East Lansing, MI 48824, USA. [3]United States Department of Agriculture, Agricultural Research Service, University of Missouri, Columbia, MO 65211, USA. [4]Department of Agronomy, University of Wisconsin, Madison, WI 53706, USA. [5]Wisconsin Crop Innovation Center, University of Wisconsin, Middleton, WI 53562, USA. [6]Department of Plant, Soil and Microbial Sciences, Michigan State University, East Lansing, MI 48824, USA. [7]Plant Resilience Institute, Michigan State University, East Lansing, MI 48824, USA. [8]Department of Statistics and Probability, Michigan State University, East Lansing, MI 48824, USA. ✉e-mail: lopezcru@msu.edu; gustavoc@msu.edu

**Table 1 | Number of records per year and region**

| Region | Year | | | | | | | | Total |
|---|---|---|---|---|---|---|---|---|---|
| | **2014** | **2015** | **2016** | **2017** | **2018** | **2019** | **2020** | **2021** | |
| North | 5306 | 5514 | 6816 | 5237 | 11,189 | 10,502 | 6476 | 8029 | 59,069 |
| South | – | 930 | 1481 | 1946 | 4344 | 3371 | 3575 | 3970 | 19,617 |
| Total | 5306 | 6444 | 8297 | 7183 | 15,533 | 13,873 | 10,051 | 11,999 | 78,686 |

**Table 2 | Number of hybrids, locations, and year-location combinations by region**

| | **North** | **South** | **Total** |
|---|---|---|---|
| Hybrids[a] | 4344 | 3013 | 4372 |
| Locations | 25 | 13 | 38 |
| Year-locations | 97 | 39 | 136 |

[a]The sum of the number of hybrids tested in the north and south is not equal to the total number of hybrids tested because many hybrids tested in northern locations were also tested in the south (see Supplementary Fig. 2).

representations[10–14]. A third and final area of renewed G×E research, particularly by private seed companies, is the combination of GBLUP models with physiological crop growth models (CGMs) in the hopes of better integrating environmental factors into genomic prediction and breeding[15–18].

In recent years, there has been tremendous growth in web resources that allow easy access to local weather and soil data. These data can be used as inputs for crop modeling to simulate crop performance under varying environmental conditions. Crop model-derived environmental covariates (EC) linked to phenotype data can be used to study how cultivars will react to environmental conditions. However, learning such patterns requires large phenotypic data sets, including diverse cultivars evaluated over many years and locations. These data sets are labor intensive to generate and, therefore, are less commonly available.

The G×E project of the Genomes-to-Fields (G2F) Initiative[19–24] has evaluated more than 4,000 maize (*Zea mays*) hybrids in multiple locations and years since 2014, resulting in a large, valuable and unique phenotypic data resource. This data set represents an opportunity for genetic investigations of cultivars' performance and stability across environments, but the size and breadth make it challenging to organize, filter, and distribute to the community in an accessible manner.

The data sets from the Maize G×E project are released with minimal formatting and data verification following a set of standard operating procedures published and revised each year of the project, as detailed in the 'description' and 'README' files in each release[19]. While the G2F organization oversees the collection of these data sets, to date, their project has not included comprehensive data cleaning, organizing, synonymizing, or any other extensive data curation efforts for all years of available data. This is done purposefully to maintain the maximum proportion of information collected by the project collaborators and allow individual researchers to define appropriate quality controls for specific projects. However, this curation strategy might limit the usability of this data set by users who are either unfamiliar with the type of data or are unable to invest the effort required to perform a more in-depth data curation prior to performing their desired analyses. Furthermore, the lack of a standardized curation process may limit reproducibility and may question the validity of comparisons between studies.

Here, we develop an automated workflow for curating the G2F genotypic and phenotypic (2014-2021) data, matching it with public weather and soil characterization data sets, and generating ECs for each year-location combination. The workflow is designed and tested using all publicly available data from the G2F trials since 2014 in the United States; however, it can be used to generate ECs for future years and historical data. This information can serve as a valuable resource to build models to predict how cultivars are expected to perform over a wide range of environmental conditions[25], including potential future climate scenarios. With this manuscript, we share this valuable resource and report on five investigations conducted using the resulting data set. First, we study the genetic and (spatial-temporal) environmental structure present in this data set. Results from this analysis provide valuable information for the design and interpretation of future studies utilizing this data set. Second, we perform analysis of variance (ANOVA) on individual ECs to determine the proportion of variance (across year-locations) attributable to region (north/south), location, and year-location interactions. Third, we study the association of individual ECs with grain yield and flowering traits using a mixed-effects model that accounts for hybrid and year-location effects. Fourth, using the EC data, we develop indices for drought and heat stress during reproductive stages and estimate the impact of such stresses on grain yield and flowering traits. This allows us to identify naturally occurring events that lead to water and heat stress during reproductive stages. Fifth, using variance components and prediction analysis (with two validation designs), we benchmark models including all the available SNPs and all the ECs and show that such models can provide moderately accurate predictions of grain yield, and highly accurate predictions of flowering traits.

## Results
### The G2F network of trials
The G2F-G×E Initiative is a collaborative testing network that has tested maize hybrids in the US and Canada since 2014. The curated phenotype data set includes 78,686 records of 4372 hybrids, tested over 8 years (from 2014 to 2021) and 38 locations in the United States (Tables 1, 2, see Supplementary Fig. 1 for a map of these locations). The number of hybrids connecting year-locations was substantial (Supplementary Fig. 2). However, it is worth noting that, as expected for a large-scale network of trials that includes a large number of hybrids, not all cultivars were tested within all year-locations within a two-year period. The data set includes plot-level phenotypic measurements of grain yield (ton/ha), days-to-anthesis, days-to-silking, and anthesis-silking interval (ASI, Supplementary Figs. 3–6 show boxplots of these traits by year, location, and region).

### The genetic structure of the hybrids tested in G2F trials
The curated genotype file includes 4372 hybrids and 98,026 SNPs. A PC analysis of an SNP-derived genomic relationship matrix[26] (**G**) reveals the structure present in this data set which includes numerous subpopulations with the clustering of hybrids being primarily associated with the (family of) tester used to develop the hybrids (Fig. 1a, Supplementary Table 1). The top-10 PCs explain roughly 50% of the variance in hybrids (Fig. 1b), providing a quantification of the strong genetic structure of the materials tested in G2F trials.

### Spatial-temporal patterns of variation in environmental covariates
We derived a set of environmental covariates specific to each year-location using the Agricultural Production Systems sIMulator (APSIM) crop model[27]. The final EC file includes 189 ECs (corresponding to the

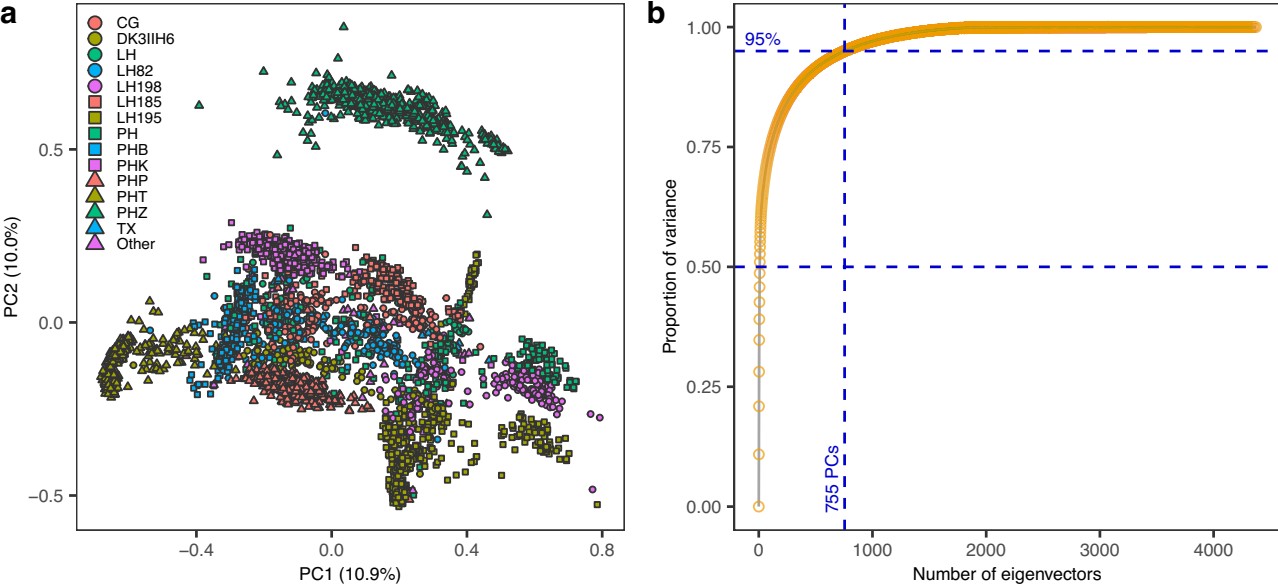

**Fig. 1 | Principal component analysis of the SNP genotypes. a** Loadings of hybrids in the first 2 SNP-derived PCs. Colors represent (groups) of testers used to develop the hybrids (Supplementary Table 1). **b** Cumulative proportion of variance of hybrids explained by the number of eigenvectors. Source data are provided as a Source Data file.

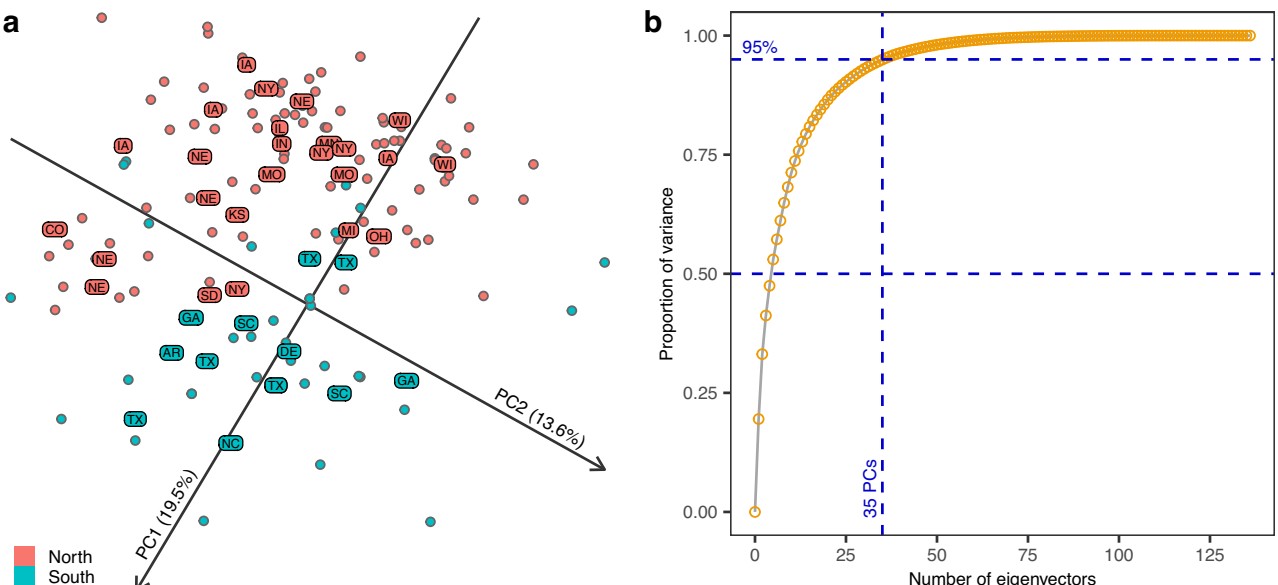

**Fig. 2 | Principal component analysis of the environmental covariates (EC). a** Loadings of the year-locations (points) in the first 2 PCs after rotating these PCs. An optimal rotation of 119 degrees clockwise was determined by maximizing the sum of the correlation of PC1 with longitude, and the correlation of PC2 with latitude. The labels correspond to the state abbreviation of locations with coordinates being the median loadings of all the year-locations with data for the corresponding location (labels for a state that appear more than once correspond to different locations of the same state). **b** Cumulative proportion of variance of ECs explained by number of eigenvectors. Source data are provided as a Source Data file.

combination of 21 EC types and 9 phenological stages, see Supplementary Tables 2, 3) evaluated in 136 unique year-location combinations. A PC analysis of these covariates showed that the top two PCs accounted for approximately 33% of the variance in the ECs (Fig. 2). Furthermore, the first two PCs effectively distinguish between northern and southern locations. To demonstrate this more clearly, we found an optimal rotation of the top two PCs that maximized the summed correlation with the longitude and latitude of the locations[28]. The optimal angle was 119 degrees clockwise. Figure 2a displays the

loadings of each year-location combination (represented by points) on rotated PCs, as well as the median loading for locations with data from multiple years (represented by labels). Overall, the rotated PCs plot shows geographic patterns, most clearly separating between northern and southern locations based only on the first two PCs. However, a separation between western and eastern locations is not easily distinguished. A few year-locations from the south are mapped to the north region and vice versa, possibly reflecting year-location effects— we study this into more detail in the next section.

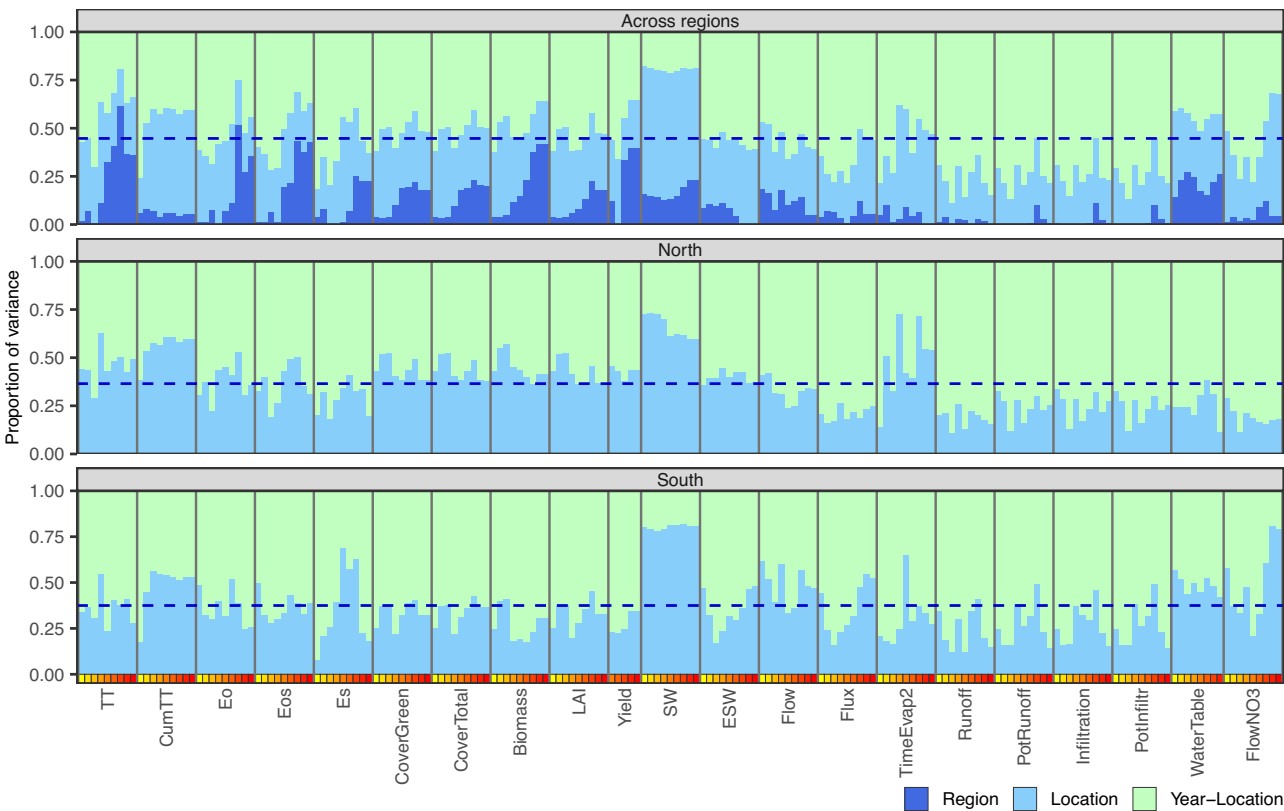

**Fig. 3 | Proportion of variance of individual environmental covariates (EC) explained by region, location, and year-location.** Each bar represents an EC type sorted by phenological stage (yellow-to-red color scale, representing stages from germination to harvest, respectively). The proportion of variance of each EC that is attributable to systematic differences between regions, locations, and year-location are separated by colors and add up to one (100%). The dashed horizontal line represents the average (across all ECs) variance explained by location. TT = Thermal time, CumTT = Cumulated thermal time, Eo = Potential evapotranspiration, Eos = Potential evaporation, Es = Realized evaporation, LAI = Leaf Area Index, SW = Soil Water, ESW = Extractable soil water, Flow = Unsaturated water movement between layers, Flux = Saturated water flux from each layer to layer below, TimeEvap2 = Time since the start of second stage evaporation, PotRunoff = Potential runoff, PotInfiltr = Potential infiltration, and FlowNo3 = Amount of Nitrogen leaching as $NO_3$ from each layer. Source data are provided as a Source Data file.

## Variability in environmental conditions explained by year-location interactions

We conducted an ANOVA of environmental covariates to quantify the proportion of variance of each EC that is attributable to systematic differences between regions, locations, and year-location interactions (see Methods for details). Results are presented in Fig. 3 for each EC-type and phenological stage. Approximately 60% of the variability in ECs is attributable to variability in weather conditions across years. The proportion of location variance was higher for thermal time (TT) and a sub-group of water ECs corresponding to soil water (SW) (Fig. 3). For many of the EC categories, the proportion of variance explained by region increased along the crop development (for example, TT and evaporation related categories), denoting cumulative environmental effects (Fig. 3).

## Association of individual environmental covariates with yield and flowering traits

We tested the association of each EC with yield and flowering traits using a mixed-effects model that includes the (fixed effect) regression on a given EC, accounting for hybrid, year-location, and hybrid-by-location interaction (random) effects (see Methods). The results of the association analysis (likelihood ratio test, LRT, followed by Bonferroni[29] correction) of ECs for northern locations are presented in Fig. 4. In the north, several ECs were significantly associated with days-to-anthesis, days-to-silking, and ASI. However, only a few environmental covariates were associated with grain yield. An enrichment analysis (EA, see red marks ·, *, **, and ***, on top of each of the plots in Fig. 4) revealed that covariates significantly associated with flowering traits were related to the cumulative thermal time (CumTT, EA, $P < 0.001$) and above-ground biomass (EA, $P = 0.0214$ for days-to-anthesis, $P = 0.0245$ for days-to-silking, and $P < 0.001$ for ASI). ECs significant associated with grain yield are related to potential evaporation (Eos, EA, $P = 0.0096$) and canopy variables (cover green, cover total, biomass, and LAI=leaf area index, EA, $P = 0.0626$). As expected, grain yield was significantly associated with APSIM's predicted yield (EA, $P < 0.05$) in late stages of the phenology. In contrast, the number of ECs significantly associated (LRT) with traits was much smaller in southern locations (Supplementary Fig. 7) compared to the north, possibly due to the smaller sample size. In the south, only evaporation-related ECs were found strongly associated with days-to-anthesis (EA, $P = 0.0099$) and days-to-silking (EA, $P = 0.0053$, Supplementary Fig. 7).

## Phenotypic effects of drought and heat stress

Using the daily outputs of the crop model we derived the water supply-demand ratio[30] (SDR) and the number of days with a maximum temperature over 30 °C (HI30) during the period between flag leaf appearance and end of grain filling (see Methods). These indices summarize, respectively, water and heat stress patterns during reproductive stages. As expected, the water SDR index reached minimum values in the period spanning from flag leaf appearance to end of grain filling in most environments (Fig. 5a). In the north region, roughly 50% of the trials had SDR < 1 during reproductive phenological stages. Whereas in the south, the prevalence of SDR < 1 was much higher (Fig. 5a). There were no clear patterns of SDR association with year, indicating that SDR

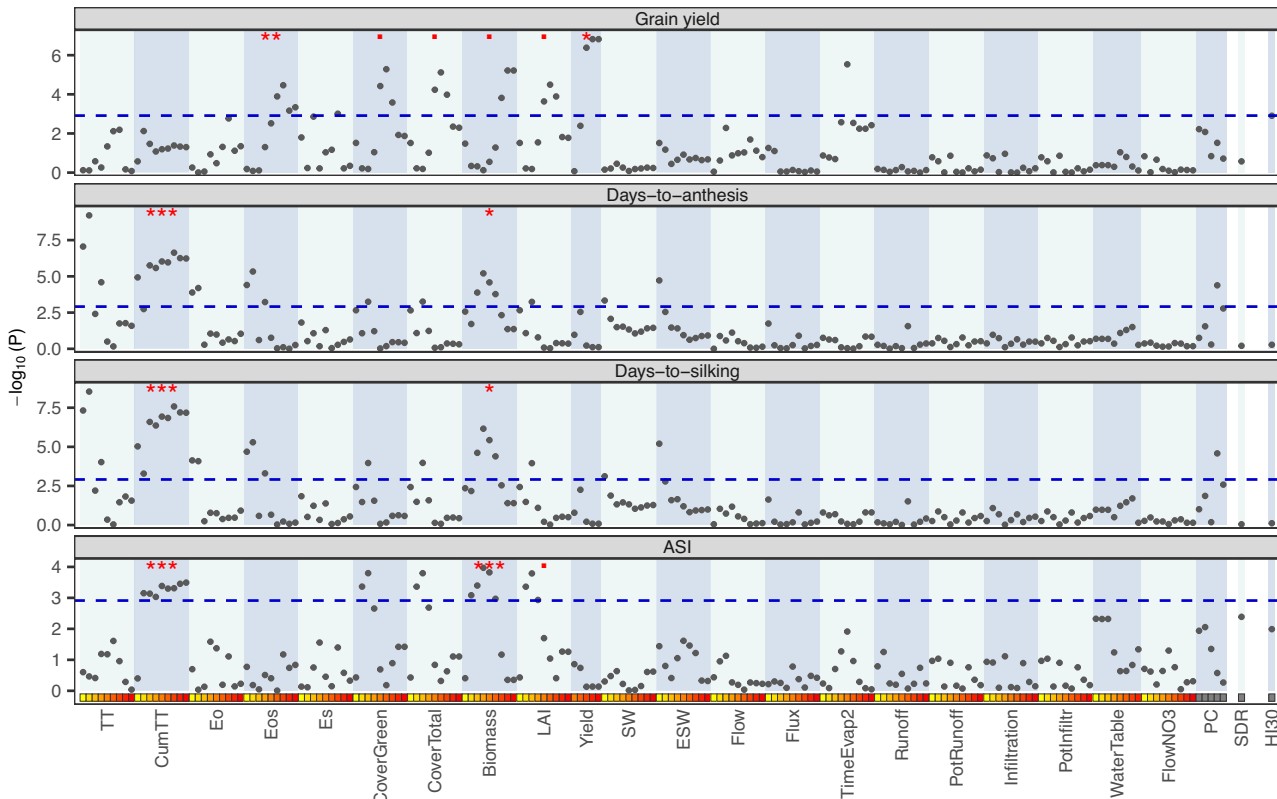

**Fig. 4 | Association between individual environmental (co)variates (EC) with yield and flowering traits in the northern trials.** $n = 59,069$ records. Each point corresponds to an EC. Association P-values (y-axis) are obtained from the likelihood ratio test (random effects model vs random effects model + EC) using a one-sided chi-square test with 1 degree of freedom. The dashed horizontal line gives the threshold for a 5% significance after Bonferroni adjustment ($P_{adj} = 0.0012$). The red marks on the top indicate whether the EC group (i.e., the covariates in the shaded vertical area) was enriched for significant associations in a one-sided Hypergeometric test ($\bullet P \leq 0.1$, $^{*}P \leq 0.05$, $^{**}P \leq 0.01$, and $^{***}P \leq 0.001$). This test was performed only for groups containing more than one covariate. TT = Thermal time, CumTT =

Cumulated thermal time, Eo = Potential evapotranspiration, Eos = Potential evaporation, Es = Realized evaporation, LAI = Leaf Area Index, SW = Soil Water, ESW = Extractable soil water, Flow = Unsaturated water movement between layers, Flux = Saturated water flux from each layer to layer below, TimeEvap2 = Time since the start of second stage evaporation, PotRunoff = Potential runoff, PotInfiltr = Potential infiltration, FlowNo3 = Amount of Nitrogen leaching as $NO_3$ from each layer, SDR = Supply-Demand ratio, HI30 = Number of days with maximum temperature over 30 °C, and PC = Top 5 PCs derived from ECs. Source data are provided as a Source Data file.

profiles vary substantially between locations within any given year (Fig. 5a).

As expected, there was a negative correlation ($r = -0.54$, $P < 0.001$) between SDR and the HI30 index. In the south, 32 of the 39 year-locations had more than 36 days with temperatures above 30 °C (denoted as HI30>36). In contrast, only 22% of the trials (22 of 97) in the north region had HI30>36. Overall, 13% of the trials (19 of the 136 year-locations) experienced both water (defined as SDR ≤ 0.54) and heat (HI30>36) stresses during reproductive stages (Fig. 5b).

The results in Fig. 4 and Supplementary Fig. 7 tested the linear association of SDR and HI30 with yield and flowering traits. The heat stress index (HI30) was marginally (linearly) associated with grain yield in the north, and the water SDR and HI30 covariates have a suggestive association with ASI; however, the P-value was above the Bonferroni-adjusted threshold (Fig. 4).

We use these ECs to define dummy variables that identify heat (HI30>36) and drought (SDR ≤ 0.54) stress. Then, we estimated (and tested) the effects of heat and drought stress using the same mixed model used in the association analysis (previous section) with the EC replaced with the dummy variables above described. In northern locations, SDR ≤ 0.54 was associated with an average reduction of grain yield of about 2.39 ton/ha (LRT, $P<0.001$), an average increase of days-to-anthesis and days-to-silking of about 3 (LRT, $P = 0.017$), and 4 days (LRT, $P = 0.011$), respectively; however, there was no clear evidence of a significant effect in ASI being associated with SDR ≤ 0.54 (Fig. 6a).

The patterns in the southern locations were similar, but showing a smaller difference between groups, for flowering traits; however, there were no significant differences in grain yield.

Likewise, in northern locations, HI30>36 was associated with a reduction of yield of ~2.17 tons/ha and increases in days-to-anthesis, days-to-silking, and ASI of 2.44, 2.97, and 0.55 days (all statistically significant, LRT, $P<0.05$), respectively (Fig. 6b). The results for HI30>36 in southern locations were not as clear as in northern locations, possibly reflecting that a large proportion of the southern locations had more than 36 days with a maximum temperature over 30 °C (Fig. 5b).

Nine of the 97 northern year-locations and 10 of the 39 southern year-locations experienced both SDR ≤ 0.54 & HI30>36 (Fig. 5b). The co-occurrence of SDR ≤ 0.54 & HI30>36 was associated with a reduction in grain yield of ~2.29 ton/ha (LRT, $P<0.001$), and an increase of days-to-anthesis of 3.6 days (LRT, $P = 0.017$) and in days-to-silking of 4.1 days (LRT, $P = 0.012$). In the south, the combined stress was reflected in smaller increases in flowering traits (0.7–1.5 days) in days-to-anthesis and days-to-silking, and ~0.8 days in ASI (Supplementary Fig. 8).

## Analysis of variance using all SNPs and ECs

We analyzed grain yield and flowering traits using two single-stage models. The first model was a standard random effects model (the one used in the association analysis without the fixed effect of the EC) that

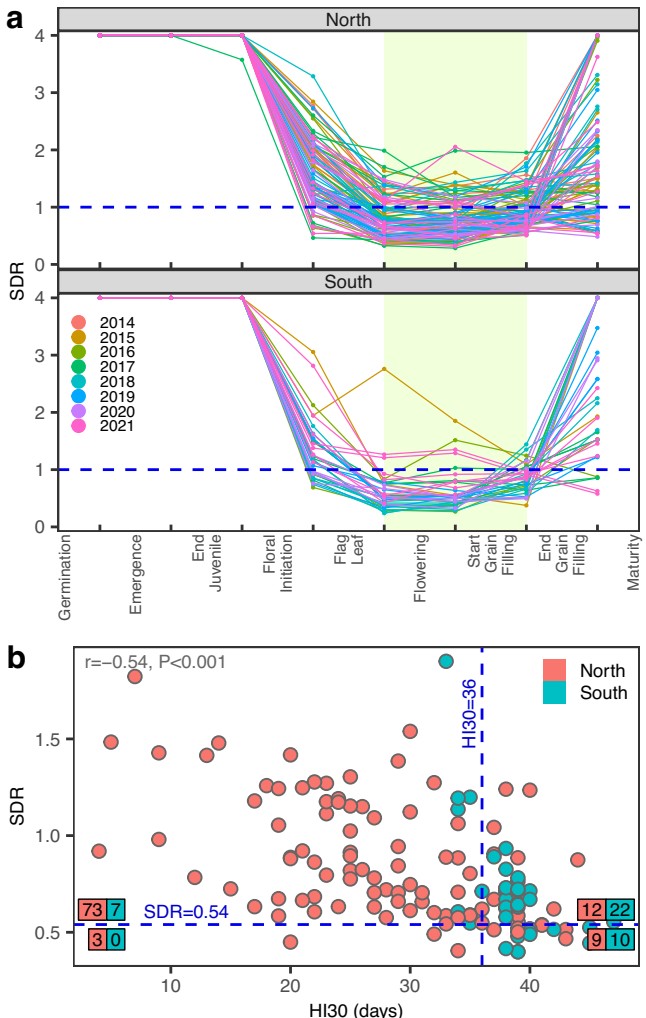

**Fig. 5 | Water supply-demand ratio (SDR) and number of days with maximum temperature over 30 °C (HI30).** **a** SDR index across phenological stages in northern and southern trials. Each line represents one year-location ($n_{YL}$ =136 year-locations) colored by year. For visualization purposes, water SDR values > 4 were set equal to 4. **b** Average SDR vs HI30 during reproductive stages (shaded area in **a**). Each point represents one year-location colored by region. The numbers represent the number of year-locations above/below the thresholds of SDR = 0.54 and HI30 = 36. *r*=Pearson correlation between SDR and HI30 followed by the *P*-value of a two-sided t-test with $n_{YL} - 2$ degrees of freedom. Source data are provided as a Source Data file.

variances estimated with a standard random effects model. Furthermore, the variance explained by SNPs was very similar to the variance captured by the main effects of the hybrids in the random effects model, and the variance captured by the main effects of the ECs was very close to the variance explained by the year, location, and year-location effects combined (Table 3 and Supplementary Table 4). Therefore, from these results, we concluded that we do not have evidence of missing heritability or missing environmentability (i.e., environmental variance not accounted for by ECs) in this data set.

**Phenotypic and genetic correlation between traits**
We estimated the phenotypic and genetic correlations between grain yield and flowering traits. For genetic correlations, we implemented a two-trait version analysis of the reaction norm model (see Methods). Grain yield had very small phenotypic and genetic correlations with anthesis and silking in the north (Supplementary Table 5). In this region, ASI had a negative phenotypic and genetic correlation with grain yield, but these correlations were also small in absolute value (~ −0.10, Supplementary Table 5). Interestingly, in the south, there was a sizable positive genetic correlation between grain yield and both anthesis and silking, and almost no phenotypic or genetic correlation between grain yield and ASI (Supplementary Table 6). As one would expect the phenotypic and genetic correlations between anthesis and silking were very close to 1 in both regions (Supplementary Tables 5, 6). Likewise, as one would also expect from the trait definition, ASI had positive (slightly negative) within-year-location phenotypic correlations with silking (anthesis) in both regions (this is a direct consequence of the trait definition, ASI=silking-anthesis). Finally, we found moderately positive genetic correlations between ASI and anthesis, and between ASI and silking in the north (Supplementary Table 5) and in the south (Supplementary Table 6). This reflects the fact that cultivars with higher growing degree days (GDD) requirement tend to have longer phases across the entire phenology, including ASI.

**Benchmark of genomic prediction models in cross-validation**
We evaluated the prediction ability of the random effects and reaction norm models using two cross-validation (CV) schemes. We first used a 10-fold CV (10F-CV) with hybrids assigned to folds. This approach (aka CV1[31]) resembles the genomic prediction of performance for (new or potential) hybrids without field evaluation. The second approach was a leave-one-year-out CV (LYO-CV) with all the data from each year assigned to testing and the data from the remaining years used for training. This approach mimics the prediction problem faced when predicting the performance of cultivars in future years with current data. We report the average within-year-location correlation between predicted and observed phenotypes for both approaches.

In the 10F-CV (hybrids assigned to folds) the average within the year-location correlation of predicted and observed phenotypes was near zero (averages between −0.10 and −0.03) for the random effects model (Fig. 7). This is expected because this model cannot learn hybrid effects because, in this CV, all the data from each cultivar is in either training or testing data. However, the model including SNPs and ECs can learn hybrid effects through genomic relationships; therefore, this model yielded considerably higher (within-year location) correlations in 10F-CV: ~0.39 (± 0.004) for grain yield, ~0.80 (± 0.002) for days-to-anthesis and days-to-silking, and ~0.33 (± 0.004) for ASI. To assess whether the prediction performance of the SNP + EC model was primarily driven by the genetic and environmental structure of the data, we considered a third model expanding the random effects model by adding the fixed effects on top PCs (top 10 PCs derived from SNPs and top 5 PCs from ECs). This random effects+PC model had a better prediction performance than the random effects model that did not use any SNP or EC information. However, the prediction performance of the PC model was significantly worse than the model using all the SNPs and ECs.

does not include any SNP or EC information. The second model is a reaction norm model[13] with a regression on SNPs and ECs only (see Methods). The results from the random effects model (without SNPs and ECs) in northern trials showed that the combined effects of year, location, and year-location interactions, explain roughly 50% of the phenotypic variance of grain yield, 85% of the phenotypic variance of days-to-anthesis and days-to-silking, and 31% of the variance of ASI (Table 3). Location effects explained between 15% (days-to-silking) to 33% (grain yield) of the variance explained by year, location, and year-location combined. Therefore, most of the variance in phenotype that can be associated with year, location, and year-location is attributable to random year and year-location effects. Within trials (year-location), hybrids explained ~0.16 for grain yield, ~0.59 for days-to-anthesis and days-to-silking, and ~0.1 for ASI (these were calculated as the ratio of the hybrid variance relative to the sum of the hybrid plus the error variance).

The estimated error variances from the reaction norm model that used SNPs and ECs were very similar (slightly smaller) to the error

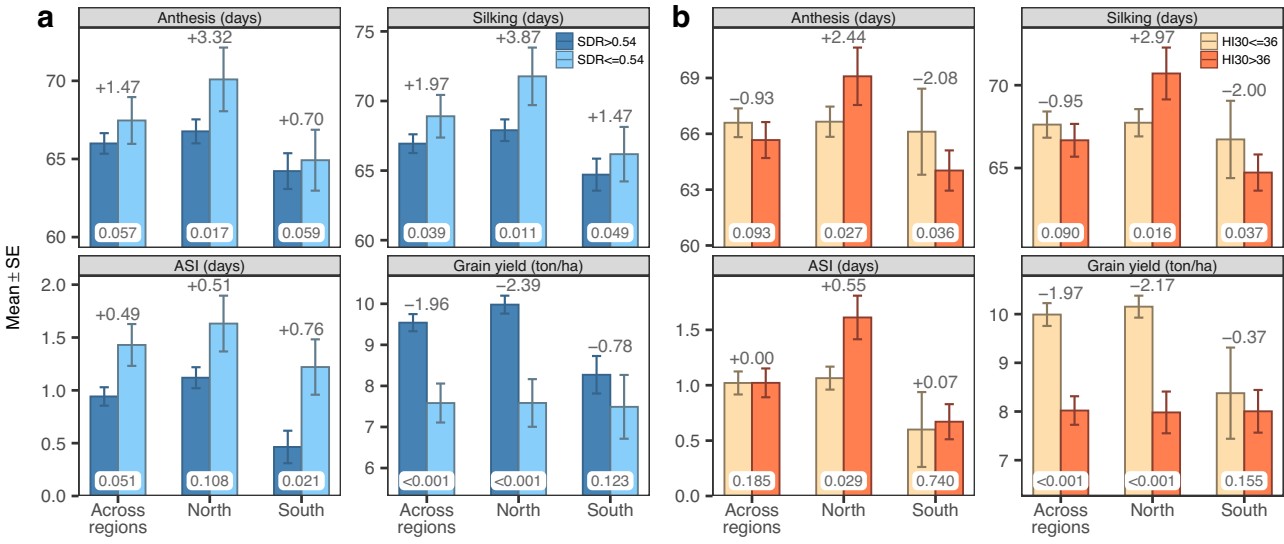

**Fig. 6 | Mean (±SE) phenotype difference between year-locations with and without stress. a** Drought stress (water supply-demand-ratio, SDR ≤ 0.54). **b** Heat stress (number of days with temperature over 30 °C, HI30>36). n = 59,069 records (north), n = 19,617 (south), n = 78,686 (across regions). Paired bars represent the contrasts intercept (left) and intercept + dummy slope (right) of the random effects model + dummy, where the dummy variable is a fixed effect for either SDR ≤ 0.54 or HI30>36. Error bars are the standard error (SE) of the contrasts. Numbers on top of bars show the mean difference between stress and no-stress given by the dummy slope. Bottom labels are the P-values of the likelihood ratio test (random effects model vs random effects model + dummy) using a one-sided chi-square test with 1 degree of freedom. Source data are provided as a Source Data file.

**Table 3 | Analysis of variance for grain yield and flowering traits in northern trials using two models**

| Source | Grain yield | | Days-to-anthesis | | Days-to-silking | | ASI | |
|---|---|---|---|---|---|---|---|---|
| | Random effects | SNP + EC | Random effects | SNP + EC | Random effects | SNP + EC | Random effects | SNP + EC |
| YEAR (Y) | 0.202 (0.042) | – | 0.273 (0.051) | – | 0.276 (0.063) | – | 0.045 (0.019) | – |
| LOC (L) | 0.168 (0.046) | – | 0.228 (0.039) | – | 0.126 (0.032) | – | 0.102 (0.034) | – |
| YL | 0.202 (0.025) | – | 0.403 (0.037) | – | 0.485 (0.036) | – | 0.214 (0.022) | – |
| (Total YL) | 0.492 (0.005) | – | 0.849 (0.004) | – | 0.840 (0.004) | – | 0.314 (0.004) | – |
| EC | – | 0.485 (0.012) | -- | 0.853 (0.009) | -- | 0.844 (0.009) | – | 0.316 (0.010) |
| Hybrid (G) | 0.079 (0.002) | – | 0.103 (0.001) | – | 0.116 (0.001) | – | 0.066 (0.002) | – |
| SNP | – | 0.067 (0.004) | – | 0.126 (0.005) | – | 0.138 (0.005) | – | 0.079 (0.006) |
| GxL | 0.041 (0.002) | – | 0.015 (0.001) | – | 0.015 (0.001) | – | 0.048 (0.003) | – |
| SNPxEC | – | 0.091 (0.004) | – | 0.045 (0.001) | – | 0.043 (0.001) | – | 0.092 (0.004) |
| Error | 0.387 (0.003) | 0.380 (0.002) | 0.074 (0.001) | 0.058 (0.001) | 0.077 (0.001) | 0.062 (0.001) | 0.566 (0.004) | 0.551 (0.003) |

All traits were scaled to a unit variance. In parenthesis, the posterior standard deviation.
*Y* Year, *L* Location, *YL* Year-location, *G* Hybrid, *EC* Environmental covariates.

The prediction correlations observed for each trait are directly related to the traits' (within-trial) heritabilities, which were highest for days-to-anthesis and days-to-silking and much lower for yield and ASI (Table 3). These differences in within-year-location prediction correlations between the random effects and SNP + EC models were highly significant for 10F-CV. The results for the southern locations were conceptually similar to those for the northern locations (Supplementary Fig. 9); however, as expected, due to the smaller sample size, the

prediction correlations were slightly lower and more variable compared to the northern locations.

In the leave-one-year-out CV (LYO-CV) analysis, all the models yielded very similar prediction correlations ranging from ~0.25-0.28 (± 0.004) for grain yield, ~0.68-0.73 (± 0.003) for days-to-anthesis and days-to-silking, and ~0.22–0.25 (± 0.004) for ASI. For days-to-anthesis and days-to-silking, the genomic SNP + EC model performed only slightly better than the random effects model that did not use SNPs and ECs (Fig. 7). These results can be explained as follows: within year-

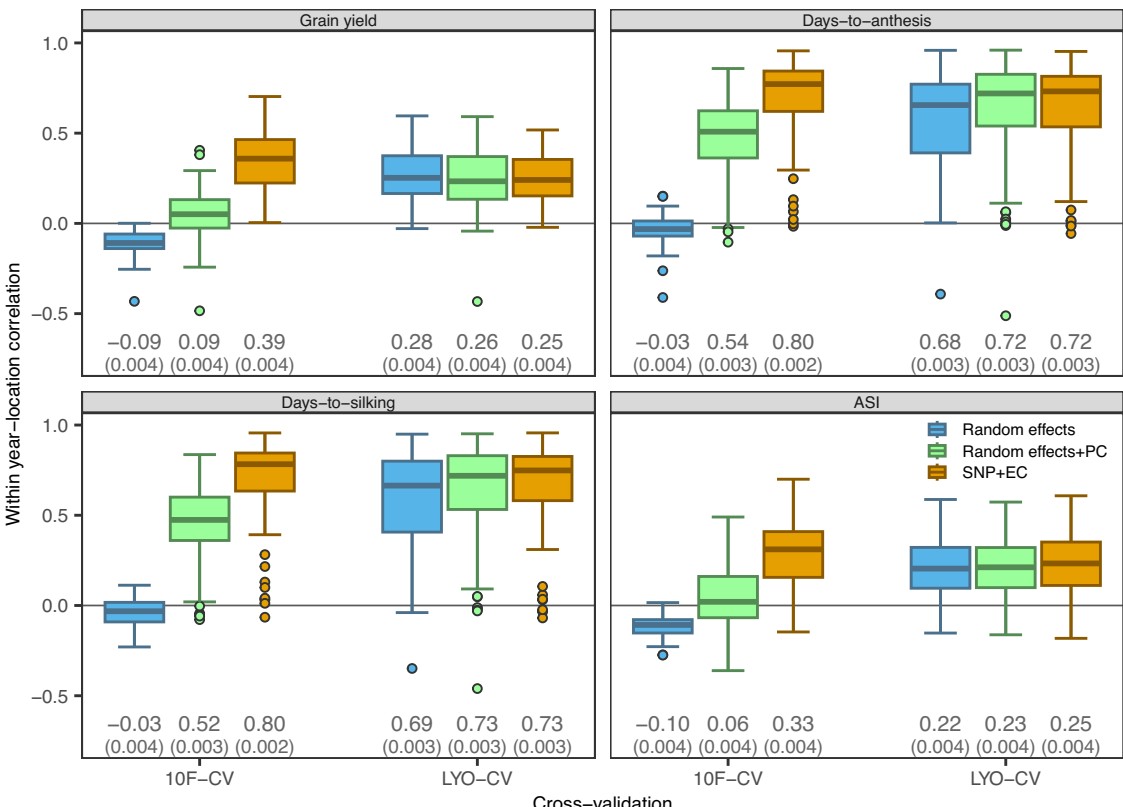

**Fig. 7 | Within year-location correlation between predicted and observed phenotypes for each model and cross-validation (CV) scheme in northern locations.** $n_{YL} = 97$ year-locations. The SNP + EC model includes SNPs and environmental covariates (EC). The random effects+PC model includes the top 10 SNPs-derived principal components (PC) plus the top 5 ECs-derived PCs. Models were fitted using 10F-CV (hybrids assigned to one of 10 folds) and LYO-CV (leave-one-

year-out). The boxes represent the inter-quartile range (IQR) bounded by the 25th and the 75th percentiles. Line at the center of each box is the median. The whiskers extend from the IQR bounds to ±1.5 times the IQR. Points represent correlations lying outside the whiskers ends. Numbers at the bottom are the weighted average correlation and standard error (SE, in parenthesis). Source data are provided as a Source Data file.

location, most of the predictable variability is attributable to differences between hybrids. Because most hybrids in the G2F data set are tested over more than one year, in this CV scheme, the random effects model can learn hybrid effects from the training data and, given the level of replication of hybrids across years and locations, the model can be as effective as the genomic model to learn those effects.

## Discussion

Leveraging data generated by the G2F Maize G×E project, web-resources that provide access to soil (SSURGO[32]) and weather (NASA POWER) information, and the APSIM crop model[27], we generated an extensive open-resource for investigating genetic, environmental, and G×E in maize in North America. The data set includes more than 70,000 phenotypic records, from more than 4000 hybrids tested in more than 130 year-location combinations, linked to curated SNP genotypes and ECs. In addition to sharing this valuable resource, we provide all the workflows that were used to curate phenotype and genotype data and to generate ECs.

Crop models rely on multiple parameters which can be optimized to match specific aspects of crop physiology to specific cultivars[33]. However, when such parameters are optimized to match observed phenotypes of specific cultivars, the resulting ECs are not purely environmental as they depend on genetic factors that affect the crop physiology. Our objective was to generate ECs that are just descriptors of the environmental conditions present in the trials; therefore, we opted not fitting a crop model with parameters to match hybrid-specific phenotypic outcomes (e.g., flowering dates by cultivar). Instead, for most crop-model parameters we used the same values across year-locations and used year-location specific values for three

parameters (planting date, plant density, and GDD to juvenile stage). Consequently, the EC data we provide and use in analyses is defined at the year-location level (i.e., there is no variability in ECs within year-location). Phenotypic prediction accuracy could be further improved by optimizing crop model parameters for specific cultivar-year-location combinations;[14, 34] however, as stated, if one does so, ECs will also be affected by genetic effects making the interpretation of genomic models (such as the SNP + EC model used here) difficult.

The G2F Maize G×E project has collected substantial environmental data (including temperature, precipitation) as well as management (e.g., fertilization) data. The incorporation of this environmental data into prediction models has shown promise when modeling G×E[6, 7, 10, 35]. However, some of this data is sparse because of missed records in some year-locations and the need to exclude specific year-locations that appear to be outliers for one or more environmental variables. In the early steps of our investigation, we used G2F environmental data and imputed the missing records with data from NASA POWER. However, this approach did not provide a clear improvement of the correlation between the year-location mean and the APSIM-predicted grain yield, over simply running the crop model with a standard fertilization, and with soil and weather data entirely downloaded from external web resources (SSURGO and NASA POWER). However, we believe there is room to further improve the resource we produced, at least for the trials with complete weather and management data, by detailed investigation of ways to incorporate weather and soil data that has been collected for some trials.

In our study we present a series of analyses describing the genetic structure of the hybrids tested in the G2F Maize G×E trials and the spatial-temporal structure of the environmental conditions that took

place at each year-location. We confirm a strong structure among the genetic materials tested in these trials (Fig. 1); such structure must be considered when performing genetic analyses with these materials. We also report that the spatial structure on the environmental conditions was largely dominated by latitude (Fig. 2a); however, close to 60% of the variability in environmental conditions was due to year and year-location (Fig. 3).

The relevance of (random) year-location effects on environmental conditions provide interesting opportunities to learn about environmental effects (as well as G × E) on the agronomic performance and stability of cultivars. For instance, one can use this data set to identify randomly occurring environmental events and use this to investigate the impact of such events on the agronomic performance of cultivars. We illustrated this by using the ECs to detect locations that may have been under drought and/or heat stress during reproductive states. Using this, we estimated a reduction in expected grain yield of about 2 ton/ha associated with the occurrence of drought stress during reproductive stages (Fig. 6). This reduction in grain yield was associated with longer days-to-anthesis and longer days-to-silking, and a slightly longer ASI. We found similar results for the effect of heat stress. A similar approach could be used to study how individual cultivars react to environmental conditions using random regression models where each cultivar has its own response to stressful conditions.

We also report on the association of individual environmental covariates with phenotypes. Our results identified several thermal-time, canopy, and water-availability ECs significantly associated with flowering traits (Fig. 4 and Supplementary Fig. 7). Some of these covariates may be capturing the same (or related) effects. Indeed, canopy ECs partially reflect water availability, radiation, and temperature. Likewise, biomass covariates integrate effects from all the conditions that contribute to the vegetative and reproductive development of the crop. In general, covariates that aggregated effects over the phenology (i.e., the ones evaluated in later stages), and those that aggregate effects across domains (e.g., canopy or biomass EC) were, as one would expect, more strongly associated with phenotypes suggesting that indeed the crop model is doing a good job at accumulating effects from various domains at different time-points. The association analysis that we presented assumed a linear relationship between the traits and ECs; future studies should explore non-linear patterns as these would be expected for many ECs.

Finally, we presented results from analyses integrating all SNPs and all ECs, including both variance components analysis as well as prediction-performance analysis. The result of the variance components analysis (Table 3 and Supplementary Table 4) suggests that SNPs fully captured hybrids effects (i.e., we found no evidence of missing heritability) and that EC data fully captured year-location effects (i.e., we had no evidence of missing environmentability). The benchmark of prediction performances confirmed that models integrating SNPs and ECs can predict cultivars' performance moderately accurately for grain yield and ASI and with high accuracy for days-to-anthesis and days-to-silking (Fig. 7 and Supplementary Fig. 9).

In a previous study involving the evaluation of wheat cultivars in France, we reported a sizable missing environmentability[25]. Some differences between the study presented by de los Campos et al.[25] and the one reported here include the crop (wheat versus maize), the crop model used, the fact that in our study ECs did not vary within year-location, and, perhaps more importantly, the diversity of environments (here we present analysis from a highly diverse set of environments within the U.S.).

The data set generated by the G2F Maize G×E project, coupled with the resources generated in this study, can enable many different types of investigations. Examples include the evaluation of models accounting for non-additive effects (here we only considered additive effects) using parametric[36–38] or semi-parametric[39, 40] methods, the use of ECs to study phenotypic plasticity, as well as the use of ECs to identify trials that may have experienced specific weather events[41]. Furthermore, the data set can be used to simulate crop performances over possible weather conditions[25]. These are just examples of countless research that could be conducted using the data and associated resources described here.

While the data generated by the G2F Maize G×E project constitutes a precious resource for maize genomic research, the data set has some limitations and specific features that need to be considered when interpreting results. First, the hybrids tested were selected to have high diversity; while this feature makes it highly valuable for many research objectives, it is worth noting that the materials tested are not representative of the elite materials currently used in commercial production. Likewise, the genetic structure of the hybrids tested is substantial and leads to a long-range linkage disequilibrium. Therefore, performing mapping requires techniques that account for such structure (e.g., within-family segregation analysis). Finally, the geographically widespread nature of the testing locations required the use of specific testers for hybrid production across subsets of locations. Therefore, testers' heterogeneity contributes to the genetic diversity of the resulting hybrids and influences genetic and G×E parameters, and models' predictive performances.

Likewise, the analyses presented here are meant to be a benchmark, where there can be many improvements to consider. For example, based on preliminary analyses (and to simplify models), we did not account for experimental design (block and replicate effects within year-location) in our models. However, the curated data set includes this information which can be used in future analyses.

The G2F Maize G×E project will continue to test materials and generate new waves of data. As new data becomes available, we aim to keep updating this valuable resource.

## Methods

### Data source

Phenotype, genotype, and metadata were downloaded from the G2F Maize G×E project public repositories[42]. These data sets consist of raw and unfiltered information from trials conducted every year from 2014 to 2021 in many locations in the United States (see Supplementary Fig. 1) and include over 4000 maize hybrids[6, 7, 22].

### Experimental design and genetic material

Hybrid phenotypic data originate from partially replicated field trials conducted across various locations in the United States, Canada, and one location in Germany. Slightly different randomization strategies were employed across the years; however, there has always been a set of common hybrids fully replicated in every location to establish a connection between locations and years (Supplementary Fig. 2).

In 2014 and 2015, based on genetic background, female parents of hybrids were classified into eight groups and crossed with up to five male testers (PB80, LH195, CG102, LH198, and LH185). Additionally, a set of common hybrids and local checks were included in the experiments, which followed a modified split-plot design[20, 22, 38].

In 2016 and 2017, along with the set of common hybrids, four subgroups of hybrids were evaluated, and the experiments were arranged in a randomized complete block design[22, 38]. The 2018-2019 experiments utilized genetic materials with a relatively narrow maturity window; the hybrids resulted from the cross between doubled-haploids (DH) from a collection of three biparental populations (PHW65 x PHN11, PHW65 x MO44, and PHW65 x MOG) to the inbred testers LH195 and PHT69. In these two years all the hybrids were evaluated in a modified randomized complete block design[23, 43].

Finally, in field seasons 2020 and 2021 the G2F Initiative evaluated hybrids produced by the cross of DH derived from the WI-SS-MAGIC population to the inbred testers PHK76, PHP02, and PHZ51. Testing was done using a modified randomized complete block design[24, 44]. Additionally, in these two years, alongside the main experiment,

smaller-scale experiments were conducted in some locations in a randomized complete block design[24]. Supplementary Table 1 provides a complete list of the testers used.

## Phenotypic and genotypic information

Phenotypic data include measurements of grain yield (ton/ha, total plot yield at 15.5% grain moisture), anthesis (days, number of days after planting that 50% of the plants in the plot began shedding pollen), silking (days, number of days after planting that 50% of the plants in the plot had visible silks), and anthesis-silking interval (ASI, difference in days between silking and anthesis). Using weather data (described below), we also expressed these flowering traits in GDD (°C-day), which is also included in the final phenotype data set. Quality control filtering included the removal of phenotypic records with at least one missing data in any of planting, harvesting, anthesis, or silking dates. We also removed yield outliers within year-location, defined as yield records greater than the 75th percentile plus 2.5 times the inter-quartile range or smaller than the 25th percentile minus 2.5 times the inter-quartile range. Likewise, we removed records with ASI greater than 15 days.

DNA genotypes were derived from a common set of 437,214 SNPs available from the Practical Haplotype Graph (PHG) platform[45]. We filtered SNPs using bcftools[46] (v1.13) by retaining bi-allelic SNPs with minor allele frequency greater than 3% and those with less than 10% of missing values. Using VCFtools[47] (v0.1.15) we coded SNP genotypes in numeric format (0,1,2) by counting the number of copies of a locus-specific reference allele. Missing values were imputed using the observed mean of marker genotypes at a given locus. Finally, we linkage disequilibrium-pruned SNPs using an algorithm that identifies sets of SNPs with $R^2>0.85$ and distance $\leq 1$ Mbp within chromosome, and that retains the most representative SNP for each set.

## Environmental covariates

Environmental covariates were derived using the Agricultural Production Systems sIMulator (APSIM Next Gen 2021.11.3.6921) crop model[27] with the 'APSIMx' R-package[48] (v2.3.1).

Most of the parameters of the crop model were set to the default values of APSIM's generic maize template. However, we tuned three key parameters to each year-location: (i) Plant density (#plants/m²) at sowing was set to the average reported plant population in each year-location. (ii) Planting date was fixed to the year-location planting date with a window of two days. (ii) We tuned the thermal time (in GDD) from emergence to end of juvenile stage (right before floral initiation) for each year-location using a grid-search approach. Specifically, we run the crop model for each year-location over a grid of values of GDD and selected the GDD value that gave a simulated flowering date equal to the average observed silking date of the year-location. Additionally, (iv) for all year-locations, the initial water was set to 50% of plant available soil water throughout the full soil profile and (v) the amount of Nitrate Nitrogen at sowing was set to 200 kg of NO₃/ha.

We run simulations starting 90 days before planting date at each year-location. Daily weather data was downloaded from the National Aeronautics and Space Administration – Prediction of Worldwide Energy Resources (NASA POWER) project between 90 days before sowing and 90 days after harvesting, and soil data was obtained from the Soil Survey Geographic Database[32] (SSURGO).

From the simulations, we saved daily values for 21 EC types (see Supplementary Table 2). APSIM provides soil-related covariates by soil layers (10 layers, each 200 mm thick). We aggregated these covariates across layers weighting for the crop accessibility to the layers. Finally, we averaged daily EC values within each phenological stage (Supplementary Table 3), thus producing 189 ECs (corresponding to the combination of 21 EC-types and 9 phenological stages).

## Region-stratified and combined analysis

We classified the testing locations into northern and southern regions according to the USDA's plant hardiness zones[49], by defining northern locations as those with hardiness 6b or below, and southern locations as those with hardiness 7a or above. This corresponds roughly to geographically dividing at the 37°N latitude. For all the analyses, we report results obtained with combined data set as well as region-stratified analyses.

## Principal component analysis

To uncover the structure of hybrids and environments in G2F data, we performed separate principal component (PC) analyses of the hybrids and ECs. Genomic PCs were derived from the eigenvalue decomposition of the SNP-derived genomic relationship matrix[26], implemented with the *eigen*() function of the 'base' R-package[50] (v4.3.1). This matrix was obtained as

$$\mathbf{G} = \frac{\mathbf{XX'}}{\text{trace}(\mathbf{XX'})} \tag{1}$$

where **X** is a matrix of centered SNP genotypes (hybrids in rows, SNPs in columns). Likewise, using ECs, we computed an environmental relationship matrix

$$\mathbf{\Omega} = \frac{\mathbf{WW'}}{\text{trace}(\mathbf{WW'})} \tag{2}$$

where **W** is a matrix of centered and scaled ECs (year-locations in rows, ECs in columns). Following Novembre et al.[28], we rotated the top two EC-derived PCs by finding an angle that maximizes the sum of the correlation of the rotated PC1 with the longitude and the correlation of the rotated PC2 with the latitude of the locations.

## Analysis of variance of environmental covariates

We used the *aov*() function of R's 'base' package[50] (v4.3.1) for the analysis of variance for each of the ECs. First, we performed ANOVA using ECs from all year-locations (south and north combined). From this analysis, we report the proportion of variance of each EC explained by region, location, and year-location. Subsequently, we performed ANOVA stratified by region. From these analyses, we report the proportion of variance explained by location and year-location interactions.

## Association of environmental covariates with phenotypes

To test the association of each EC with yield and flowering traits, we used a single-stage mixed-effects model using individual plot data taking care of the field replication, this model is of the form

$$y_{ijkl} = \mu + G_i + Y_j + L_k + YL_{jk} + GL_{ik} + EC_{jk}\beta + \varepsilon_{ijkl} \tag{3}$$

Above, $y_{ijkl}$ is a phenotypic record of the $l^{th}$ replicate, of the $i^{th}$ hybrid, in the $j^{th}$ year, in the $k^{th}$ location; $\mu$ is an intercept; $G_i$, $Y_j$, $L_k$, and $YL_{jk}$ are the random effects of the hybrid, year, location, and year-location combination, respectively; $GL_{ik}$ is a hybrid-by-location interaction (also treated as random), $EC_{jk}\beta$ is a (fixed-effect) regression on a given EC (separate analyses were performed for each EC), and $\varepsilon_{ijkl}$ is the random error term. In this model, the effects of the levels of each of the random effects were assumed to be independently and identically distributed (iid) Gaussian, that is $G_i \overset{iid}{\sim} N(0,\sigma^2_{hyb}), Y_j \overset{iid}{\sim} N(0,\sigma^2_{year}), L_k \overset{iid}{\sim} N(0,\sigma^2_{loc})$, and $YL_{jk} \overset{iid}{\sim} N(0,\sigma^2_{year \times loc})$, and $GL_{ik} \overset{iid}{\sim} N(0,\sigma^2_{hyb \times loc})$. The error term was also assumed to be iid Gaussian, $\varepsilon_{ijkl} \overset{iid}{\sim} N(0,\sigma^2_{\varepsilon})$.

We fitted the above model using the 'lme4' R-package[51] (v1.1.34) and used the fitted models to determine the association of each EC with phenotypes using a likelihood ratio test between model in Eq. (3) and a null model that had the same random effects but did not include the regression on the EC. This test was implemented using the *anova*() function from the 'base' R-package[50] (v4.3.1). From these analyses we report association *P*-values for each EC and phenotype.

We also considered including a hybrid-by-year interaction in Eq. (3); however, we could not fit these models with 'lme4' and when we fitted model in Eq. (3) with the 'BGLR' R-package[52] (v1.1.0), the hybrid-by-year interaction captured a very small proportion of variance (<2%). Therefore, in our analysis we only included the hybrid-by-location term as this can be, from the prediction perspective, the most relevant systematic and predictable interaction.

Likewise, Eq. (3) does not include random effects for the blocks and replicates within year-location. We opted not to include those effects because block information was sparse and because, in preliminary analyses using 'lme4', we found that the replicate (within year-location) explained less than 1% of the variance of the phenotypes with only the exception of grain yield in the north, where that random effect explained 4% of the variance of the trait.

We performed the above-described analysis using data combined across regions as well as within region. To control for multiple testing, we used the Bonferroni method[29]. We performed PC analysis of the ECs matrix to estimate the number of independent tests[53] and implemented in the 'poolr' R-package[54] (v1.1.1). The estimated number of independent tests was 41; therefore, our threshold for significance at 5% for these analyses was 0.05/41 = 0.00122.

### Evaluation of drought and heat stress during reproductive stages

We used the environmental data to compute an index for the water supply-demand ratio[30] (SDR). The crop water supply (mm) was the weighted sum of the water available within each soil layer that is reached by the crop at any given time. The weights were based on the fraction of the water that the crop model estimates can be extracted by the crop (which depends on root growth and soil properties). The water demand corresponds to the amount of water (mm) the crop would have transpired without soil water constraints. This was computed from the potential evapotranspiration (Eo) adjusted by the proportion of green canopy cover times a crop factor equal to 2. We then produced an overall SDR value per year-location by averaging the daily SDR observed within the period comprised between the flag leaf appearance and end of grain filling stages (see Supplementary Table 3). Likewise, for the same reproductive stages, we counted the number of days with a maximum temperature over 30 °C, and we labeled the resulting index as HI30.

First, we tested the linear association of these indices using the model in Eq. (3). Subsequently, we used the above indices to identify year-location combinations under drought and heat stress and estimated the impact of that stress on phenotypes using a mixed-effects model as in Eq. (3) with the regression on the ECs replaced with an indicator variable for presence/absence of heat or drought stress. We used a likelihood ratio test between this model and a null model without the indicator variable. This test was implemented using the *lrtest*() function from the 'lmtest' R-package[55] (v0.9.40).

### Genomic prediction models

We analyzed yield and flowering traits using two single-stage models. The first (baseline) model was a standard random effects model at the plot-level such as the one in Eq. (3) without the fixed effect of the EC, that is,

$$y_{ijkl} = \mu + G_i + Y_j + L_k + YL_{jk} + GL_{ik} + \varepsilon_{ijkl}. \tag{4}$$

From these analyses, we report the proportion of variance explained by hybrid, year, location, year-location, and hybrid-by-location interactions.

This model in Eq. (4) does not include any SNP or EC information. Therefore, our second model replaced the hybrid, year, location, year-location, and hybrid-by-location effects with a regression on SNPs and ECs. To implement this approach, we used the reaction norm model of Jarquín et al.[13] which takes the following form

$$y_{ijkl} = \mu + g_i + E_{jk} + gE_{ijk} + \varepsilon_{ijkl} \tag{5}$$

where, as before, $y_{ijkl}$ is the phenotypic record of the $l^{th}$ replicate of the $i^{th}$ hybrid in year-location $jk$; and $g_i$, $E_{jk}$, and $gE_{ijk}$ are random effects representing regressions on SNPs, ECs, and SNP-by-EC interactions, respectively. For these random effects, we assumed $\mathbf{g} \sim \mathrm{MVN}(\mathbf{0}, \sigma^2_{snp}\mathbf{G})$, $\mathbf{E} \sim \mathrm{MVN}(\mathbf{0}, \sigma^2_{ec}\mathbf{\Omega})$, and $\mathbf{gE} \sim \mathrm{MVN}(\mathbf{0}, \sigma^2_{snp \times ec}\mathbf{K})$, where $\mathbf{G}$ is the SNP-derived genomic relationship matrix[26] (Eq. 1), $\mathbf{\Omega}$ is the (linear) environmental relationship matrix derived from the ECs (Eq. 2), and

$$\mathbf{K} = (\mathbf{Z_1 G Z'_1}) \circ (\mathbf{Z_2 \Omega Z'_2}) \tag{6}$$

is a (co)variance structured derived by taking the Hadamard product ('∘') of the genomic relationships and environmental relationships. Here $\mathbf{Z_1}$ and $\mathbf{Z_2}$ are incidence matrices linking phenotypes with hybrids, and year-location combinations, respectively.

The genetic materials tested in G2F trials have a relatively strong population structure. Likewise, there is also a spatial structure on environmental conditions largely associated with latitude. Therefore, to assess whether differences in the predictive performances of models in Eqs. (4) and (5) could be explained by genetic or spatial (environmental) structure, we considered a third model which expanded the random effects model (Eq. 4) by adding the fixed effects of top SNP- and EC-derived PCs. We included the top-10 PCs from SNPs and top-5 PCs from ECs (denoted as $V_{ij}$ and $U_{jks}$, respectively), each explaining ~ 50% of the SNPs and ECs, respectively:

$$y_{ijkl} = \mu + G_i + Y_j + L_k + YL_{jk} + GL_{ik} + \sum_{r=1}^{10} V_{ij}b_r + \sum_{s=1}^{5} U_{jks}d_s + \varepsilon_{ijkl}. \tag{7}$$

We implemented models in Eqs. (4), (5), and (7) using the 'BGLR' R-package[52] (v1.1.0). In this data set sample size is larger than the number of predictors; therefore, for models in Eqs. (4), (5), and (7) we recommend using the *BLRXy*() function from the 'BGLR' R-package which works using sufficient statistics optimizing computations for problems with $n \gg p$[56]. The computational bottleneck for models in Eq. (5) is the factorization (e.g., eigenvalue decomposition) of the Hadamard product matrix $\mathbf{K}$ in Eq. (6). For this task, instead of factorizing $\mathbf{K}$ using the R's *eigen*() function, we developed an algorithm that derives a basis for $\mathbf{K}$ from the eigenvectors of $\mathbf{G}$ and $\mathbf{\Omega}$ (this tool is also included with this manuscript).

### Phenotypic and genetic correlation between traits

Although most of our analyses were based on single-trait models, we also performed two-trait analyses from where we report phenotypic and genetic correlations between grain yield and flowering traits.

The phenotypic correlations between traits were computed across-, within-, and between-year-locations. Across year-locations, phenotypic correlations were computed using Pearson's correlation coefficients (*r*) between traits within each region (i.e., across all trials). These correlations are largely influenced by differences in the year-location means. Therefore, we also report estimates of phenotypic correlations within- and between-year-locations. To estimate the within-year-location phenotypic correlation between traits, we computed Pearson's correlation between traits within each year-location and then computed a weighted average, with weights given by the

standard error (SE) of the correlation,

$$SE(r_i) = \sqrt{\frac{1 - r_i^2}{n_i - 2}} \qquad (8)$$

where $n_i$ is the number of records in the $i^{th}$ year-location. Finally, to estimate the between-year-locations phenotypic correlation between traits, we estimated the year-location means for each trait using model in Eq. (4) and then computed Pearson's correlation between the year-location means of each pair of traits.

For genetic and environmental correlations, we used the *Multi-trait*() function of the 'BGLR' R-package[52] (v1.1.0) to fit a two-trait version of model in Eq. (5). From this model, we report estimates of the genetic correlations between traits as well as environmental correlations. Model in Eq. (5) has two environmental terms, the regression on EC and the error terms. The EC capture differences between year-locations and the error term captures within-year-location environmental effects. Therefore, we report separate estimates of the environmental correlations for each of these terms.

### Evaluation of prediction performance

To evaluate the prediction ability of our models and data, we performed two cross-validation (CV) analyses. Following Burgueño et al.[31], in a first CV scenario (10F-CV), we randomly assigned hybrids to 10 non-overlapping folds. We predicted all the records of hybrids in the $k^{th}$ fold using a model trained with all records from hybrids in the remaining folds. In this scenario, we are mimicking the problem of predicting the performance of new hybrids that have not undergone any field evaluation.

The second approach is a leave-one-year-out CV (LYO-CV) approach. Thus, in this setting, we predicted all data for a given year using all the data from the remaining years for model training. These analyses resemble the prediction problem faced when predicting the performance of cultivars in future years with current data.

We evaluated the prediction ability as the Pearson's correlation between observed plot-level phenotypes and predictions from models in Eqs. (4), (5), and (7). A significant portion of the variance in grain yield and flowering traits can be attributed to year-location effects. As a result, models that can capture year-location means may achieve higher prediction correlations, even if they do not perform well at ranking hybrids within year-location combinations. However, from a plant breeding perspective, the ability to rank hybrids within environments is more important. Therefore, we primarily compared models based on their average within-year-location correlation between predicted and observed phenotypes.

### Reporting summary

Further information on research design is available in the Nature Portfolio Reporting Summary linked to this article.

## Data availability

All the phenotypic, agronomic, and metadata considered in this study are released every year and are available under Public Domain Dedication on the G2F website [https://www.genomes2fields.org/resources]. Genotypic data was obtained from CyVerse Data Store [https://doi.org/10.25739/tq5e-ak26][57]. Soil data was sourced from the Soil Survey Geographic Database [https://sdmdataaccess.nrcs.usda.gov][32]. Weather data was downloaded from the NASA Langley Research Center (LaRC) POWER Project [https://power.larc.nasa.gov]. The aggregated curated data set (including the SNP genotypes, phenotypes, and ECs) is available in the Figshare repository [https://doi.org/10.6084/m9.figshare.22776806][58]. Source data are provided with this paper.

## Code availability

The scripts used to implement all the analyses described in this study are provided in the GitHub repository [https://github.com/QuantGen/MAIZE-HUB].

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

## Acknowledgements

The authors would like to thank all the G2F collaborators that every year contribute to generate this wonderful resource. We also thank James Holland and Anna Rogers who contributed to initial discussions about the need to generate the comprehensive resource we are presenting in

this article. M.L.C., N.D.L., S.M.K., L.W.D.L.B., A.M.T., R.T., and G.D.L.C. acknowledge financial support from NSF PGRP-Tech grant #2035472. M.L.C. and G.D.L.C. received support from USDA grant #67015-33413, and from Michigan State University. J.D.W., N.D.L., and S.K. acknowledge the support from the United States Department of Agriculture—Agricultural Research Service. A.M.T. and R.T. received support from the Michigan State University—Plant Resilience Institute.

## Author contributions

G.D.L.C., N.D.L., and S.M.K. conceived the idea. M.L.C., F.A., J.W., A.M.T., R.T., and G.D.L.C. contributed to develop the crop model pipeline. D.C.L., L.W.D.L.B., and N.D.L. worked with G2F collaborators to perform field trials experiments and to collect phenotypic and genotypic data. All authors participated in writing and preparing this manuscript.

## Competing interests

The authors declare no competing interests.
