## [Peer Review File · Nature Communications]

Leveraging Data from the Genomes-to-Fields Initiative to Investigate G×E in Maize in North AmericaReviewers' Comments:

Reviewer #1:

Remarks to the Author:

The authors are proposing an analysis workflow for the G2F dataset in which they apply certain statistical analysis and include environmental covariates associated with weather, soil and others derived from a crop model. This is an important effort at investigating GxE interactions.

This appears to be a valuable contribution, especially for scientists involved in the G2F project. I appreciate the efforts the authors have put into making the data and code available.

I would have appreciated a short review of similar attempt at analyzing large datasets for GxE interactions in the context of plant breeding.

I would suggest re-writing the first paragraph of the introduction with a more clear focus. I would have expected a more emphasis on the general challenge of GxE interactions, especially in plant breeding. For example, the second sentence brings up soil and weather databases, but this contribution is more notable for the phenotypic data. Also, I would like to know about similar efforts if any.

In this set of 5 objectives, there is inclusion of partial results. Like any reviewer, I'm biased, but I do not think this structure works well for this manuscript.

LN 125. Which version of APSIM are you using?

I'm on line 128 and I read that 'only a few parameters were calibrated'. I would certainly like to know which parameters and how. 'NO3 Nitrogen' is slightly confusing. I'm guessing that it is better to omit the 'NO3'.

LN 248-249. I don't think reading the data into R belongs in the main text. Many readers would enjoy seeing details of the analysis in the Supplementary material. I suggest including this information using that format.

Table 2. I would include in the caption some hint about why does North + South = Total for Locations and Year-Locations, but not for hybrids. Is there that much overlap for hybrids between N and S?

LN 408-417. Discussion might come below, but it is interesting to speculate about the many reasons why it is easier to model flowering than yield.

LN 453-454. Do you mean that you manually adjusted GDDs in APSIM to match the observed silking date? Or did you use an optimization method?

LN 497-499. It looks like this analysis is not able to explore interactions among predictors and nonlinear relationships. This is probably the reason for the relatively low predictive ability for yield and ASI.

Reviewer #2:

Remarks to the Author:

Multi environment field trials are very important in plant breeding but currently existing data often suffer from one or more of the following problems: The trials are very unbalanced, the data not publicly available, small number of environments and environmental covariates that are either lacking or of low quality. The present paper describes a data set with in total 138 environments and a large

number of hybrids; necessarily not completely balanced but with most hybrids present in at least two years.

I think this is a paper many people were waiting for, and is likely to be widely cited. Nevertheless I have some major and minor concerns, in particular related to the question if the data set will really be a benchmark for future research. The size (though important) should not be the only consideration here, but also the quality of the data, and whether the data set has certain properties that earlier data sets were lacking. Other concerns are the experimental designs of the individual trials that are currently not described.

Please note that I did not have time to review earlier papers on the g2f project, so if the authors believe that a certain point can be addressed by simply referring to earlier work that is fine in principle. At the same time certain information would be nice to include to make the paper more self-contained.

Major comments:

(1) as far as I can see no information whatsoever is provided about the experimental design of the original field trials, while at the same time models (1-4) are apparently at the plot level; the index l refers to the l th replicate. This is actually nice in my opinion because it allows for separation of GxE interactions and residual error. It seems the authors estimated genotypic effects (BLUEs?) in each trial, using some kind of mixed model, and then took the sum of these genotypic effects and residuals, effectively treating each trial as a completely random design in subsequent analysis. Please explain if this is the case, or, if not, how design effects were accounted for. I had a quick look at the R scripts but did not see it there (admittedly I did not have time to run the scripts or inspect them line by line).

(2) supplementary figures S1 to S5 visualize each trait separately but nowhere the relation between yield and flowering time is reported; what about phenotypic and genetic correlations within trials, or what about plotting environmental main effects for different traits against each other? And finally to which extent there was confounding between location and the choice of hybrids grown there? (due to flowering time). variation in flowering time is of course necessary in such a large geographical region, and of interest in its own right. it would be helpful if the reader would get some idea about the relation between yield and anthesis /silking, e.g. how much genetic variation approximately is unique for yield ?

(3) lines 394 to 400: These appear to be the most remarkable findings of the paper. Although I completely trust the authors, it almost sounds too good to be true, at least that was my first thought. Over the last years I have looked at several datasets using this SNP+ EC model with BGLR, including subsets of the genome to fields data, and always found a substantial amount of missing 'environmentability' (20-40%, say) and even more missing gxe; sometimes also some missing heritability (e.g. due to dominance). Ruling out coding errors or statistical issues, it is especially here that I'm wondering what precisely makes this data set different. I hope the authors can clarify this using other data from the literature or by analyzing subsets of the present data. Is it simply the size of the data that makes this possible? In that case one would expect some missing variation in case the analysis is restricted to random subsets of hybrids and environments. Another possibility is that most variation in yield is driven by flowering time. In that case I would expect some missing environmentability in case the analysis is restricted to genotypes within a given flowering window, either medium early or late. By the way I think it would be helpful if that information could be provided for each hybrid, in case it's not in the files already.

Minor comments:

- Model 1 in line 166, as well as models 2 and 4 later on: please motivate the absence of the genotype by year interaction. Was it found to be consistently small or does it have to do with one of the cross

validations schemes defined at the end, in which each fold corresponds to a year?

- Line 200: I agree that such binary variables are useful to describe certain stress scenarios but still I was wondering if the SDR or HI30 themselves could not be used in model 1.

- Line 224: this model considers 5 principal components for the genetic as well as the environmental part, while in order to explain the same amount of variance in both parts, far more components are needed for the genetic part. Moreover, why there are no PC terms for the interaction? Is this a computational issue?

- Lines 236 to 242: was accuracy evaluated by computing these correlations from plot or replicate level data or correlations between observed genotypic means and predicted genetic effects? The latter option is most relevant I believe.

- Figure 4: What about also considering SDR and HI30 as well as the first principal components as covariates here? These would appear as single points because they integrate different covariates from the nine stages; it would be of interest to see how much more significant they would be. By contrast I did not understand the yield covariate, especially for the top panel about grain yield itself. If it would be relevant to analyze one trait conditioning on another, I would expect anthesis and silking as covariates.

- Related to the previous point, lines 358 to 377: It should become clear here that the SDR and HI30 indices are more relevant than individual covariates measured at individual time points. I do not doubt that this is the case but not directly clear because this part focuses on effect sizes whereas the part about Figure 4 focuses on significance. Both are relevant but a more explicit comparison between the individual covariates and the indices would be helpful.

- Line 429 to 431: what really seems to matter is where you are in Figure 6, which appears to define the most relevant stress scenarios. Most often, it seems, all the years in the training set adequately sample from this two-dimensional space, and this is how you would like it to be as a plant breeder, i.e. all important scenarios are represented in the training set at least a few times. In any case all locations are by construction included in the training set.

-For readers with a interest in genomic prediction it would be worth mentioning how the authors managed to run bglr for such a large data set

Reviewer #3:

Remarks to the Author:

This paper reports really excellent work summarizing an enormous data set (for public sector crop science programs) in a nicely streamlined way, laying out a number of clear analysis questions and their results. I have very little to criticize, the work is done very well and the writing is excellent.

Small suggestions are to cite these papers that are relevant to the topics and discussion:

Rogers et al 2021 <https://doi.org/10.1093/g3journal/jkaa050>

Rogers and Holland 2022 <https://doi.org/10.1093/g3journal/jkab440>

These papers focus on only the first three years of this data set, and their use of summarized weather data by period windows is very likely inferior to what the current manuscript has done using APSIM. It might be good to show that the current work builds on and improves this earlier work. The 2021 paper also shows the importance of fitting heterozygosity-based relationships in this data set, presumably due to the capture of dominance effects. It's hard to see how the current models could be improved much if there is not missing heritability, but that previous paper shows a clear improvement of the

matrix that captures dominance as well as additive effects over the pure additive model.

Millet et al 2019 <https://doi.org/10.1038/s41588-019-0414-y>

This paper describes improved environment-specific prediction based on crop models tuned to each genotype using previous phenotype data available on all genotypes (including the test set). Of course, that is problematic from the pure prediction point of view, but it does show the potential for crop model-based predictions (perhaps the parameters of the model can be predicted from pure genomic data). You allude to this idea in discussion section, I think you should cite this paper at that point.

The hyperlink to the code is not working in the supplemental file, so I can't figure out how to access that code. People are going to want to be able to implement these ideas and will need the code.

Dear Editors and Reviewers,

We thank you very much for the useful comments provided.

In response to your comments, we have revised the manuscript and provide below a point-by-point response to each of your comments. The text that was modified/added in the manuscript is highlighted in **red font**. This response letter includes the lines containing the modifications done in response to your comments.

Thank you very much again for your comments.

REVIEWER COMMENTS

Reviewer #1 (Remarks to the Author):

The authors are proposing an analysis workflow for the G2F dataset in which they apply certain statistical analysis and include environmental covariates associated with weather, soil and others derived from a crop model. This is an important effort at investigating GxE interactions.

This appears to be a valuable contribution, especially for scientists involved in the G2F project. I appreciate the efforts the authors have put into making the data and code available.

R1: Thank you very much for appreciating our work.

I would have appreciated a short review of similar attempt at analyzing large datasets for GxE interactions in the context of plant breeding.

I would suggest re-writing the first paragraph of the introduction with a more clear focus. I would have expected a more emphasis on the general challenge of GxE interactions, especially in plant breeding. For example, the second sentence brings up soil and weather databases, but this contribution is more notable for the phenotypic data. Also, I would like to know about similar efforts if any.

R2: We followed your suggestion and revised the first paragraph in the Introduction section. The revised version offers a short review of previous studies that have analyzed GxE using different approaches. See **lines 49-64**.

In this set of 5 objectives, there is inclusion of partial results. Like any reviewer, I'm biased, but I do not think this structure works well for this manuscript.

R3: Thank you very much for your comment. We understand that style varies between fields and journals. We included these highlights because we thought that such remarks will engage readers. We will consult with the editor about his opinion and make a final decision on whether to remove these remarks based on the editor's recommendation.

LN 125. Which version of APSIM are you using?

R4: We used APSIM Next Gen version 2021.6.4.6515—we added this information in **lines 164-165**.

I'm on line 128 and I read that 'only a few parameters were calibrated'. I would certainly like to know

which parameters and how. 'NO3 Nitrogen' is slightly confusing. I'm guessing that it is better to omit the 'NO3'.

R5: Thank you very much for this suggestion. We now provide detailed information about which parameters had year-location-specific values and how these parameters were tuned (see **lines 166-175**). We also edited the reference to Nitrogen (see **lines 174-175**).

LN 248-249. I don't think reading the data into R belongs in the main text. Many readers would enjoy seeing details of the analysis in the Supplementary material. I suggest including this information using that format.

R6: We removed this from the main text.

Table 2. I would include in the caption some hint about why does North + South = Total for Locations and Year-Locations, but not for hybrids. Is there that much overlap for hybrids between N and S?

R7: Done. See the footnote that we added to Table 2. See also **lines 327-328** commenting this, and heatmap (Supplementary Figure S2) with the number of hybrids in common between locations.

LN 408-417. Discussion might come below, but it is interesting to speculate about the many reasons why it is easier to model flowering than yield.

R8: Our results are in line with previous studies (e.g., Buckler, et al., Science, 2009) that have shown that Maize flowering traits are highly heritable and have a relatively simple (compared with grain yield) genetic architecture which is mainly additive.

LN 453-454. Do you mean that you manually adjusted GDDs in APSIM to match the observed silking date? Or did you use an optimization method?

R9: We used a grid-search optimization procedure (see R5 above and text in **lines 166-175**). We also edited the text that you refer to make it clearer (see **lines 557-559**)

LN 497-499. It looks like this analysis is not able to explore interactions among predictors and nonlinear relationships. This is probably the reason for the relatively low predictive ability for yield and ASI.

R10: Some of our models accounted for by-linear interactions between SNPs and EC (see eq. [3]). Furthermore, we partially disagree with your assessment about the predictive ability for grain yield and ASI. These are highly complex traits, the prediction correlations (which are defined within year location) are, in our view, good. Furthermore, there has not been consistent empirical evidence suggesting that non-linear models could lead to a significant increase in prediction accuracy for GxE models.

Reviewer #2 (Remarks to the Author):

Multi environment field trials are very important in plant breeding but currently existing data often suffer from one or more of the following problems: The trials are very unbalanced, the data not publicly available, small number of environments and environmental covariates that are either lacking or of low quality. The present paper describes a data set with in total 138 environments and

a large number of hybrids; necessarily not completely balanced but with most hybrids present in at least two years.

I think this is a paper many people were waiting for, and is likely to be widely cited.

R11: Thank you very much for appreciating the value of our manuscript.

Nevertheless I have some major and minor concerns, in particular related to the question if the data set will really be a benchmark for future research. The size (though important) should not be the only consideration here, but also the quality of the data, and whether the data set has certain properties that earlier data sets were lacking.

R12: This is a very important suggestion. We added a paragraph in the discussion highlighting the two most notable limitations that we found for this data set (see **lines 629-639**).

Other concerns are the experimental designs of the individual trials that are currently not described. Please note that I did not have time to review earlier papers on the g2f project, so if the authors believe that a certain point can be addressed by simply referring to earlier work that is fine in principle. At the same time certain information would be nice to include to make the paper more self-contained.

R13: Thank you for this suggestion. We added an entire section where we summarize the different designs used over the years (new section added starting in **line 121 through 143**).

Major comments:

(1) as far as I can see no information whatsoever is provided about the experimental design of the original field trials, while at the same time models (1-4) are apparently at the plot level; the index *l* refers to the *l*th replicate. This is actually nice in my opinion because it allows for separation of GxE interactions and residual error.

R14: You are correct, our model accounts for replicates within trial (see also R13 above).

It seems the authors estimated genotypic effects (BLUEs?) in each trial, using some kind of mixed model, and then took the sum of these genotypic effects and residuals, effectively treating each trial as a completely random design in subsequent analysis. *Please explain if this is the case, or, if not, how design effects were accounted for. I had a quick look at the R scripts but did not see it there (admittedly I did not have time to run the scripts or inspect them line by line).*

R15: We used one-stage models where genetic and environmental (either specific aspects of the design or EC) were modeled jointly. This is described in models (1)-(4) which, as you stated, are defined at the plot level. To avoid any misunderstanding, we now state explicitly (**lines 207-208, 251, 262**) that we use single-stage models and edited the description of models [1] through [4] to make it explicit that in all models the response was an individual phenotypic record, not means derived from another model.

(2) Supplementary figures S1 to S5 visualize each trait separately but nowhere the relation between yield and flowering time is reported; what about phenotypic and genetic correlations within trials, or what about plotting environmental main effects for different traits against each other?

R16: Thank you very much for this important suggestion. In response to your comment, we expanded our analysis and now also report estimates of phenotypic and genetic correlations between traits. The methodology used is described in **lines 285-304** and the results are presented in the added **Table 4**, text in **lines 492-504**, and added **Supplementary Table S5**. The scripts used for these analyses are also included in the GitHub repository of the manuscript: (<https://github.com/QuantGen/MAIZE-HUB/>).

And finally to which extent there was confounding between location and the choice of hybrids grown there? (due to flowering time). variation in flowering time is of course necessary in such a large geographical region, and of interest in its own right.

R17: The experiments are highly connected even between the northern and southern locations (see Supplementary Figure S2 and R7 above). However, even knowing that the connections are very high, we still decided to report separate estimates for the south and north, throughout the manuscript because these are very different growing regions.

It would be helpful if the reader would get some idea about the relation between yield and anthesis /silking, e.g. how much genetic variation approximately is unique for yield?

R18: See R16 and Table 4. Briefly, our estimates of the genetic and environmental correlations between Yield and anthesis and yield and silking were very close to zero. There was however, a negative environmental correlation between year location means (also captured by EC), but this correlation was small in absolute value (~ -0.1).

(3) lines 394 to 400: These appear to be the most remarkable findings of the paper. Although I completely trust the authors, it almost sounds too good to be true, at least that was my first thought. Over the last years I have looked at several datasets using this SNP+ EC model with BGLR, including subsets of the genome to fields data, and always found a substantial amount of missing 'environmentability' (20-40%, say) and even more missing gxe; sometimes also some missing heritability (e.g. due to dominance). Ruling out coding errors or statistical issues, it is especially here that I'm wondering what precisely makes this data set different. I hope the authors can clarify this using other data from the literature or by analyzing subsets of the present data. Is it simply the size of the data that makes this possible? In that case one would expect some missing variation in case the analysis is restricted to random subsets of hybrids and environments. Another possibility is that most variation in yield is driven by flowering time. In that case I would expect some missing environmentability in case the analysis is restricted to genotypes within a given flowering window, either medium early or late. By the way I think it would be helpful if that information could be provided for each hybrid, in case it's not in the files already.

R18: To assess whether there is missing environmentability we have compared the variance accounted for by year-location in the random effects model (2) and the variance captured by ECs in model (3). For this assessment it is essential that model (3) does not include year, location, or year-location effects; therefore, in this model the environmental variance is not 'divided' between EC and year, location, and year-location effects. To compare with previous studies, we need to focus on those that have used similar models. For example, de los Campos et al. (Nat. Comm., 2020) use similar models for the study of GxE in wheat in France. This study, reported a sizable "missing environmentability" by which we mean that the EC captured only a fraction of the year-location (year+location+year-by-location) variance.

Important differences between de los Campos et al. (Nat. Comm., 2020) and the study we are presenting include: (i) the crop (de los Campos et al. Nat. Comm., 2020 study wheat, our study focuses in Maize), (ii) the crop model used (we used APSIM, de los Campos et al., 2020 used a custom model used by ARVALIS, (iii) the diversity of environments (the study we are presenting is much more diverse in terms of environments), and (iv) the fact that EC in our study were defined at the year-location level (previous studies also optimized parameters for each cultivar-year-location combination). We believe that the fact that we are dealing with a different crop and, perhaps more importantly, with a much more diverse set of environments may explain the differences between studies in this respect. However, this is clearly an area that deserves further research. We added a reference to the above-mentioned study in **lines 615-619**.

Minor comments:

- Model 1 in line 166, as well as models 2 and 4 later on: please motivate the absence of the genotype by year interaction. Was it found to be consistently small or does it have to do with one of the cross validations schemes defined at the end, in which each fold corresponds to a year?

R20: With the sizable spatial diversity of environments the scope of year effects that are systematic across locations is limited. Furthermore, most cultivars are tested over two years; thus, we don't have a systematic evaluation of cultivar performance over many years. Finally, from prediction perspective, the most relevant interaction (the one that can be systematic and predictable and leveraged to assort cultivar to locations) is genotype-by-location because future year conditions are unknown. These were the reasons we considered when we decided not to include genotype-by-year interactions. However, in response to your comment we evaluated the inclusion of hybrid-by-year interactions. We could not fit those models using lme4 (lme4 won't fit models with a number of levels of random effects larger than sample size) and when we fitted the models using BGLR, as one would expect for the reasons above-stated, the proportion of variance explained by the hybrid-by-year interaction was negligible (<2%). This information is now provided in **lines 221-226**.

- Line 200: I agree that such binary variables are useful to describe certain stress scenarios but still I was wondering if the SDR or HI30 themselves could not be used in model 1.

R21: Good suggestion. In the revised submission we now include these variables in the linear association analysis (see sentence in **line 245**, and Figures 4 and S7, **lines 402-405** for results).

- Line 224: this model considers 5 principal components for the genetic as well as the environmental part, while in order to explain the same amount of variance in both parts, far more components are needed for the genetic part. Moreover, why there are no PC terms for the interaction? Is this a computational issue?

R22: We agree with your suggestion of using a number of PCs that explain a similar proportion of variance; therefore, we run again model (4) including 10 PCs derived from SNPs and 5 PCs from ECs (see edited text in **lines 274-276**). All the results were updated using 10PCs for SNPs.

We did not include PCs from the interaction between SNPs and EC because, upon inspection, the top PC for the interaction terms did not show any apparent structure not captured by EC and SNP PCs—this is a direct consequence of the strong replication of genotypes over year-locations (see added Supplementary Figure S2).

- Lines 236 to 242: was accuracy evaluated by computing these correlations from plot or replicate

level data or correlations between observed genotypic means and predicted genetic effects? The latter option is most relevant I believe.

R23: The correlations we report are the average of the within-year-location correlations between plot-level phenotypes with predictions (this is now highlighted in **line 315**). Because we computed correlations within year-locations, these correlations are not inflated by systematic differences due to year-location means.

- Figure 4: What about also considering SDR and HI30 as well as the first principal components as covariates here? These would appear as single points because they integrate different covariates from the nine stages; it would be of interest to see how much more significant they would be. By contrast I did not understand the yield covariate, especially for the top panel about grain yield itself. If it would be relevant to analyze one trait conditioning on another, I would expect anthesis and silking as covariates.

R24: We included SDR and H130 in the results in Figure 4 and Figure S7. Note that here we are testing linear association of SDR with individual covariates. In the section that follows Figure 4 we use SDR to define a drought stress covariate and testing non-linear associations with phenotypes, including combined effects of drought and heat stress.

The yield covariate is the APSIM predicted yield for each of the stages (we make it explicit in **lines 398-399**).

- Related to the previous point, lines 358 to 377: It should become clear here that the SDR and HI30 indices are more relevant than individual covariates measured at individual time points. I do not doubt that this is the case but not directly clear because this part focuses on effect sizes whereas the part about Figure 4 focuses on significance. Both are relevant but a more explicit comparison between the individual covariates and the indices would be helpful.

R25: Thank you for the suggestion. The analyses reported in Figure 4 (and Figure S7) test linear associations between quantitative covariates and phenotypes. In the section that follows Figure 4, we look more closely at the effects of drought and heat stress. To that end, we use the quantitative covariates (SDR and the number of days with maximum temperature over 30° C degrees) to define dummy variables that identify trials under heat and or drought stress (see Figure 6). The results in Figure 7 test associations between these dummy variables and phenotypes. We clarify the difference between the linear association analysis and the one based on dummy variables that define heat and drought stress in **lines 442-446**.

- Line 429 to 431: what really seems to matter is where you are in Figure 6, which appears to define the most relevant stress scenarios. Most often, it seems, all the years in the training set adequately sample from this two-dimensional space, and this is how you would like it to be as a plant breeder, i.e. all important scenarios are represented in the training set at least a few times. In any case all locations are by construction included in the training set.

R26: In the two CV schemes that we considered (CV1 and CV2) the training data includes a mix of trials with and without stress, this happens because, as you noted, usually in most years there are locations that suffer some level of heat and drought stress.

-For readers with a interest in genomic prediction it would be worth mentioning how the authors managed to run bglr for such a large data set

R27: Thank you for the suggestion. We provide more details on the ‘tricks’ used to fit these models in **lines 279-284**. All the scripts and tools we used are included in the GitHub repository that we created for this manuscript (<https://github.com/QuantGen/MAIZE-HUB>).

Reviewer #3 (Remarks to the Author):

This paper reports really excellent work summarizing an enormous data set (for public sector crop science programs) in a nicely streamlined way, laying out a number of clear analysis questions and their results. I have very little to criticize, the work is done very well and the writing is excellent.

R28: Thank you very much for your comments.

Small suggestions are to cite these papers that are relevant to the topics and discussion:

Rogers et al 2021 <https://doi.org/10.1093/g3journal/jkaa050>

Rogers and Holland 2022 <https://doi.org/10.1093/g3journal/jkab440>

These papers focus on only the first three years of this data set, and their use of summarized weather data by period windows is very likely inferior to what the current manuscript has done using APSIM. It might be good to show that the current work builds on and improves this earlier work. The 2021 paper also shows the importance of fitting heterozygosity-based relationships in this data set, presumably due to the capture of dominance effects. It's hard to see how the current models could be improved much if there is not missing heritability, but that previous paper shows a clear improvement of the matrix that captures dominance as well as additive effects over the pure additive model.

Millet et al 2019 <https://doi.org/10.1038/s41588-019-0414-y>

This paper describes improved environment-specific prediction based on crop models tuned to each genotype using previous phenotype data available on all genotypes (including the test set). Of course, that is problematic from the pure prediction point of view, but it does show the potential for crop model-based predictions (perhaps the parameters of the model can be predicted from pure genomic data). You allude to this idea in discussion section, I think you should cite this paper at that point.

R29: Thank you for these references we were missing. We included these references in the revised manuscript (see Experimental Design section and **line 562**, sentence in **lines 566-568, and 623**)

The hyperlink to the code is not working in the supplemental file, so I can't figure out how to access that code. People are going to want to be able to implement these ideas and will need the code.

R30: Thank you. We deposited our workflow in the following GitHub repository:
<https://github.com/QuantGen/MAIZE-HUB>

Reviewers' Comments:

Reviewer #1:

Remarks to the Author:

The authors have addressed the comments

Reviewer #2:

Remarks to the Author:

The authors have done a lot of extra work, and I really appreciate all the additional analysis (especially regarding SDR + hi30, and the genetic correlations between yield and anth./silking). I am also grateful for the new things about BGLR that I learned. I recommend publication because the rebuttal is very satisfactory on most points. Only for the following mostly minor issues i'm not entirely convinced. They are certainly not an obstacle for publication

R13-15, related to experimental design. The authors now give a nice overview of the different designs and emphasize they did a one stage analysis, but it is still not clear if and how design effects were actually accounted for. If they were simply ignored this would be useful to know so future work might potentially improve on this. If the data were corrected for design effects in earlier works please also mention this. Finally if they were included in the present analysis please mention this in models one through four (in words, or byincluding some generic design term)

I completely agree with R23 but if I'm not mistaken, within year/location correlations can still be inflated or deflated when calculated at plot level, depending on the heritability, nb of replicates and hybrids, genomic prediction method used etc.

I guess that for the present manuscript this will make very little difference for the comparison between models, although it might become an issue when future works want to compare accuracies.

Reviewer #3:

Remarks to the Author:

Authors have responded to my comments adequately. I am not sure their updated description of field experimental designs are sufficiently informative. The designs were complex and varied among years, but it might be useful to indicate that, due to the large number of entries, not all entries were tested within all environments within a 2-year period. In addition, there were two levels of blocking within environments: "replications" and "blocks within replications", labelled as "Replicate" and "Block" in the trait data file, but "Replicate" blocks were generally not complete replicates. I note that neither of these two factors were included in their analysis model, maybe their effects are not important?

Dear Editors and Reviewers,

We thank you very much for the useful comments provided.

In response to your comments, we have revised the manuscript and provide below a point-by-point response to each of your comments. The text that was modified/added in the manuscript is highlighted in **red font**. This response letter includes the lines containing the modifications done in response to your comments.

Thank you very much again for your comments.

REVIEWER COMMENTS

Reviewer #1 (Remarks to the Author):

The authors have addressed the comments

R1: Thank you very much for reviewing our manuscript and for your previous comments, we found them very useful.

Reviewer #2 (Remarks to the Author):

The authors have done a lot of extra work, and I really appreciate all the additional analysis (especially regarding SDR + hi30, and the genetic correlations between yield and anth./silking). I am also grateful for the new things about BGLR that I learned. I recommend publication because the rebuttal is very satisfactory on most points.

R2: Thank you very much !

Only for the following mostly minor issues i'm not entirely convinced. They are certainly not an obstacle for publication.

R13-15, related to experimental design. The authors now give a nice overview of the different designs and emphasize they did a one stage analysis, but it is still not clear if and how design effects were actually accounted for. If they were simply ignored this would be useful to know so future work might potentially improve on this. If the data were corrected for design effects in earlier works please also mention this. Finally if they were included in the present analysis please mention this in models one through four (in words, or by including some generic design term)

R3: You are correct in that we did not include replicate (within year location) and block effects in the models. We did not include those effects for a few reasons: (i) block IDs were not always available (many had NAs in the source data), (iii) mixed effects models including year, location, year-location, and replicate-year-location effects did not converge. This was likely due to the small replicate effect. To see that, we used lmer to fit mixed-effects models with year-location combined (instead of year+location+year-location, a model that did not converge) and also included in the model the replicate-year-location effects. These analyses revealed that replicate (within year-location)

explained less than 1% of the variance of the phenotypes (with only one exception, ~ 4% for yield in the north). The following table shows the estimated variance components by trait and region obtained with lmer.

Table 1: Variance components analysis from a mixed-effect model with year-location, genotype, genotype-location, and year-loc-replicate as random effects (fitted with lmer). The phenotype was standardized to unit variance; therefore, variance estimates can be interpreted as proportion of variance explained.

Region	Source	Grain yield	Anthesis	Silking	ASI
North	Year-location	0.5717	0.8974	0.9022	0.3421
	Hybrid	0.0726	0.0901	0.0983	0.0628
	Hybrid-location	0.0420	0.0153	0.0157	0.0484
	Rep (year-loc)	0.0407	0.0126	0.0109	0.0104
	Error	0.3760	0.0663	0.0702	0.5599
South	Year-location	0.7024	0.9192	0.9079	0.3971
	Hybrid	0.0667	0.0698	0.0659	0.0287
	Hybrid-location	0.0379	0.0103	0.0115	0.0562
	Rep (year-loc)	0.0170	0.0107	0.0115	0.0082
	Error	0.2983	0.0611	0.0721	0.5577

In response to your comments, we made the following edits to our manuscript:

- We provide in the curated data set the replicate and block information in case other scientists want to use it for their analysis
- We include in the methods and the discussion text that explains that we did not model block and replicate effects and mention this as something that could be done differently in future analysis (see lines 227-231 and 650-653).

I completely agree with R23 but if I'm not mistaken, within year/location correlations can still be inflated or deflated when calculated at plot level, depending on the heritability, nb of replicates and hybrids, genomic prediction method used etc.

I guess that for the present manuscript this will make very little difference for the comparison between models, although it might become an issue when future works want to compare accuracies.

R4: The correlation estimates should be nearly unbiased. In some studies people report correlations divided by the square-root of heritability. In this case, accuracies are expressed in terms of the maximum correlation that could be achieved if knew the true genetic values because the correlation between phenotypes and genetic values is the square-root of the trait heritability. We opted not to do this because averaging relative accuracies would give higher weights to trials with very low heritability. Thus, while $Cor(\hat{y}, y) / \sqrt{h^2}$ may be a good metric in studies involving just one trial, we did not find it adequate for our study.

Reviewer #3 (Remarks to the Author):

Authors have responded to my comments adequately. I am not sure their updated description of field experimental designs are sufficiently informative. The designs were complex and varied among years, but it might be useful to indicate that, due to the large number of entries, not all entries were

tested within all environments within a 2-year period. In addition, there were two levels of blocking within environments: "replications" and "blocks within replications", labelled as "Replicate" and "Block" in the trait data file, but "Replicate" blocks were generally not complete replicates. I note that neither of these two factors were included in their analysis model, maybe their effects are not important?

R5: We edited the description of the experimental designs adding some of the information you mention (see lines 332-334). Regarding replicate and block effects, we refer you to R3 above.

Reviewers' Comments:

Reviewer #2:

Remarks to the Author:

Thank you very much for the clarifications about the designs; I recommend publication

Reviewer #3:

Remarks to the Author:

Authors have responded adequately to previous comments, this seems ready for publication. Nice work!